# ⚎ MUSE: Machine Unlearning Six-Way Evaluation for Language Models

**Weijia Shi**[*1]   **Jaechan Lee**[*1]   **Yangsibo Huang**[*2] **Sadhika Malladi**[2]   **Jieyu Zhao**[3]
**Ari Holtzman**[4]   **Daogao Liu**[1]   **Luke Zettlemoyer**[1]   **Noah A. Smith**[1]   **Chiyuan Zhang**[5]
[1]University of Washington   [2]Princeton University
[3]University of Southern California   [4]University of Chicago   [5]Google Research
https://muse-bench.github.io

## ABSTRACT

Language models (LMs) are trained on vast amounts of text data, which may include private and copyrighted content, and data owners may request the removal of their data from a trained model due to privacy or copyright concerns. However, exactly unlearning only these datapoints (i.e., retraining with the data removed) is intractable in modern-day models, leading to the development of many approximate unlearning algorithms. Evaluation of the efficacy of these algorithms has traditionally been narrow in scope, failing to precisely quantify the success and practicality of the algorithm from the perspectives of both the model deployers and the data owners. We address this issue by proposing **MUSE**, a comprehensive machine unlearning evaluation benchmark that enumerates six diverse desirable properties for unlearned models: (1) no verbatim memorization, (2) no knowledge memorization, (3) no privacy leakage, (4) utility preservation on data not intended for removal, (5) scalability with respect to the size of removal requests, and (6) sustainability over sequential unlearning requests. Using these criteria, we benchmark how effectively eight popular unlearning algorithms on 7B-parameter LMs can unlearn Harry Potter books and news articles. Our results demonstrate that most algorithms can prevent verbatim memorization and knowledge memorization to varying degrees, but only one algorithm does not lead to severe privacy leakage. Furthermore, existing algorithms fail to meet deployer's expectations, because they often degrade general model utility and also cannot sustainably accommodate successive unlearning requests or large-scale content removal. Our findings identify key issues with the practicality of existing unlearning algorithms on language models, and we release our benchmark to facilitate further evaluations.

## 1 INTRODUCTION

Training language models (LMs) often involves using vast amounts of text data, which may inadvertently contain private and copyrighted content (Carlini et al., 2021; Henderson et al., 2023; Min et al., 2023; He et al., 2024). In real-world applications, data owners may demand that their data be removed from a trained language model due to privacy or copyright concerns, as mandated for example by the General Data Protection Regulation (GDPR, European Parliament & Council of the European Union). Moreover, recent copyright lawsuits (*DOE 1 v. GitHub, Inc.*, N.D. Cal. 2022; *Tremblay v. OpenAI, Inc.,*, 2023) emphasize the need for removing copyrighted data from the model.

These recent developments have intensified research interest in designing, evaluating, and improving *machine unlearning* algorithms, which aim to transform an existing trained model into one that behaves as though it had never been trained on certain data (Ginart et al., 2019; Liu et al., 2020; Wu et al., 2020; Bourtoule et al., 2021; Izzo et al., 2021; Gupta et al., 2021; Sekhari et al., 2021; Ye et al., 2022b; Ghazi et al., 2023). Exact unlearning in LMs requires removing the undesired data (the *forget set*) and retraining the model from scratch on the remaining data (the *retain set*), which is too costly to be practical, especially for frequent unlearning operations. As such, several efficient approximate unlearning algorithms have been proposed (Eldan & Russinovich, 2023; Zhang et al., 2024b), but existing evaluations of LM unlearning on question answering (Eldan & Russinovich, 2023; Maini et al., 2024) cannot provide a holistic view of how practical and effective a particular

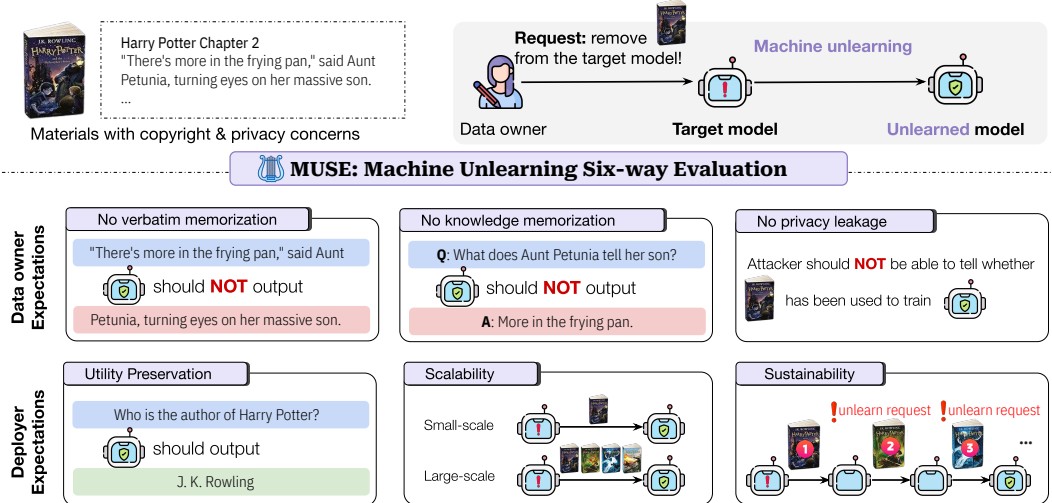

Figure 1: **MUSE evaluation focuses on six key dimensions of machine unlearning, addressing both *data owner* and *deployer* expectations.** For example, when an author (data owner) requests the unlearning of the Harry Potter books, they may expect the unlearned model to: (1) avoid generating verbatim copies of the text to protect copyright, (2) eliminate retention of factual knowledge from the books, and (3) not reveal whether the books were previously used in training to protect privacy. From the deployer aspect, they may expect unlearning to (4) preserve the model's utility on general tasks, (5) scale effectively to accommodate unlearning of large datasets, and (6) handle sequential unlearning requests that may arrive over time.

unlearning algorithm is. In this work, we propose a systematic, multi-faceted framework called **MUSE** (**M**achine **U**nlearning **S**ix-Way **E**valuation; §3) to evaluate six desired properties for unlearning algorithms (Figure 1). Our criteria cover both the data owner's and the model deployer's desiderata for a practical unlearning algorithm. Data owners require the LM to unlearn the precise tokens (*verbatim memorization*), general knowledge encoded in the tokens (*knowledge memorization*), and any indication that their data was included in the training set to begin with (*privacy leakage*). On the other hand, model deployers want to effectively accommodate many successive unlearning requests (*sustainability*) on various sizes of forget sets (*scalability*) without degrading the general model capabilities (*utility preservation*).

We apply **MUSE** to evaluate **eight representative machine unlearning algorithms** (§4) on **two datasets** (§3.2), focusing on the specific cases of unlearning Harry Potter books and news articles. Our findings indicate that most unlearning algorithms remove verbatim memorization and knowledge memorization with varying degrees of efficacy but operate at the cost of utility preservation and do not effectively prevent privacy leakage (§5.2). In particular, negative preference optimization (NPO; Zhang et al., 2024b) and task vectors (Ilharco et al., 2023) are especially effective in removing these types of memorization, but we find that NPO often permits privacy leakage and both methods induce a sharp drop in the utility of the model. Furthermore, testing their scalability and sustainability reveals that they both algorithms struggle with large forget sets and successive unlearning requests (§5.3).

Our results highlight that unlearning algorithms generally fail to meet data owner expectations in preventing privacy leakage, which is one of the primary motivations for unlearning. Additionally, they struggle to meet all three of the aforementioned deployer expectations. Therefore, although it is increasingly desirable to find an efficient and effective unlearning algorithm amid rising concerns around privacy regulations and copyright litigations, our evaluation suggests that currently feasible unlearning methods are not yet ready for meaningful usage or deployment in real-world scenarios. These findings underscore the pressing need for further research in this area. We also release our benchmark to facilitate further evaluations and welcome extensions to other modalities.

## 2 MACHINE UNLEARNING: PRELIMINARIES AND NOTATIONS

Machine unlearning (Ginart et al., 2019; Liu et al., 2020; Izzo et al., 2021; Sekhari et al., 2021; Gupta et al., 2021; Ye et al., 2022b; Liu et al., 2024) has emerged as an important capability to accommodate data removal requirements that arise from scenarios with privacy or copyright concerns.

Table 1: **Comparison with a previous benchmark**: Unlike the previous benchmark TOFU (Maini et al., 2024), which evaluates unlearning on synthetic Q&A datasets, **MUSE** tackles real-world unlearning challenges: unlearning real-world large-scale corpus ($22\times$ larger) while taking into account six desiderata that are important to both data owners and deployers. More related works are discussed in Appendix 6.

|  |  | **MUSE** (ours) | TOFU (Maini et al., 2024) |
|---|---|:---:|:---:|
| **Evaluation criteria** | C1. No verbatim memorization | ✓ |  |
|  | C2. No knowledge memorization | ✓ | ✓ |
|  | C3. No privacy leakage | ✓ |  |
|  | C4. Utility preservation | ✓ | ✓ |
|  | C5. Scalability | ✓ |  |
|  | C6. Sustainability | ✓ |  |
| **Evaluation corpora** | Domains | NEWS and BOOKS | Synthetic autobiographies |
|  | Data Constitution | Verbatim text and knowledge set (Q & A) | Q & A |
|  | Scale (# tokens in forget set) | 0.8M for NEWS, 3.3M for BOOKS | 0.15M |

We briefly describe the machine unlearning setting. Consider a dataset $\mathcal{D}_{\text{train}}$ and a model $f_{\text{target}}$ trained on $\mathcal{D}_{\text{train}}$. Suppose we design an algorithm $\mathcal{U}$ to unlearn a specific subset (i.e., the *forget set*) $\mathcal{D}_{\text{forget}} \subset \mathcal{D}_{\text{train}}$ from $f_{\text{target}}$. We want to preserve performance on a *retain set* $\mathcal{D}_{\text{retain}} = \mathcal{D}_{\text{train}} \setminus \mathcal{D}_{\text{forget}}$, and we also evaluate the model on an in-distribution but disjoint *hold-out set* $\mathcal{D}_{\text{holdout}}$ which the model has never been trained on. So, the unlearning algorithm $\mathcal{U}$ takes $f_{\text{target}}$, $\mathcal{D}_{\text{forget}}$, and, optionally, $\mathcal{D}_{\text{retain}}$ and outputs an unlearned model $f_{\text{unlearn}}$. Exact unlearning ensures $f_{\text{unlearn}}$ is behaviorally identical to the model resulting from retraining from scratch, denoted $f_{\text{target}}$, but such retraining is usually too costly in real world deployment, so we focus on evaluating approximate unlearning algorithms.

## 3 THE **MUSE** EVALUATION BENCHMARK

**MUSE** evaluates a comprehensive set of desirable properties of machine unlearning across six facets. We detail the evaluation metrics in §3.1 and describe the evaluation corpus in §3.2.

### 3.1 EVALUATION METRICS

Ideally, an unlearned model should behave as if it had never seen the forget set, exhibiting similar behavior to a retrained model on any corpus $\mathcal{D}$ such that $m(f_{\text{unlearn}}, \mathcal{D}) \approx m(f_{\text{retrain}}, \mathcal{D})$, where $m$ represents any evaluation metric. Prior evaluations on LM unlearning focus on performance of specific tasks like question answering (e.g., Eldan & Russinovich, 2023; Maini et al., 2024). However, these metrics do not faithfully reflect data owner expectations and real-world deployment considerations when performing unlearning. To address this, we propose comprehensive evaluation metrics that consider both *data owner* and *deployer* expectations. A comparison between **MUSE** and the prior benchmark is shown in Table 3.

**Data owner expectations.** When removing a forget set from a model, data owners typically have three main expectations regarding the unlearned model: (C1) **No verbatim memorization**: The model should not exactly replicate any details from the forget set. (C2) **No knowledge memorization**: The model should be incapable of responding to questions about the forget set. (C3) **No privacy leakage**: It should be impossible to detect that the model was ever trained on the forget set. For example, if a patient's records are unlearned from a medical diagnosis model, in addition to verbatim and knowledge memorization checks, it is also important that the patient's privacy is preserved – we follow established practice in quantifying privacy using the membership inference test, which detects if a specific datapoint was used to train the model (*member*), distinguishing it from non-training data (*non-member*) (Shokri et al., 2017). In this case of unlearning a record from a diagnostic model, it is undesirable for the model to leak membership information, because it would be used to associate the patient with the disease. We quantify these data owner expectations with three evaluation metrics:

**C1. No verbatim memorization** When a model has unlearned a medical record, it should not output its contents verbatim. We quantify the verbatim memorization VerbMem by prompting the model with the first $l$ tokens from a sequence $x_{[:l]} \in \mathcal{D}_{\text{forget}}$ and comparing the continuation outputted by the model $f$ to the true continuation $x_{[l+1:]} \in \mathcal{D}_{\text{forget}}$ using the ROUGE-L F1 score (Lin, 2004).

$$\text{VerbMem}(f, \mathcal{D}) := \frac{1}{|\mathcal{D}_{\text{forget}}|} \sum_{x \in \mathcal{D}_{\text{forget}}} \text{ROUGE}(f(x_{[:l]}), x_{[l+1:]})$$

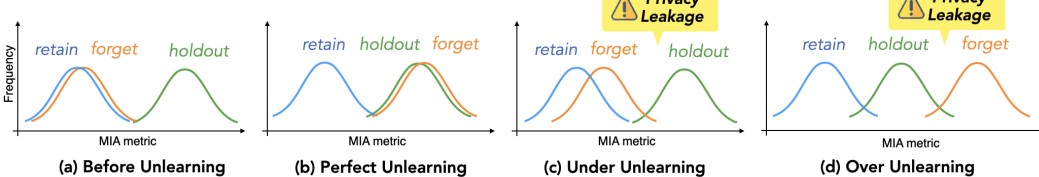

Figure 2: Distribution of the MIA metric (see C3) for $\mathcal{D}_{\text{forget}}$, $\mathcal{D}_{\text{holdout}}$, and $\mathcal{D}_{\text{retain}}$. **Differences in the metric between forget and holdout sets indicate various unlearning outcomes of $\mathcal{D}_{\textbf{forget}}$, potentially leaking privacy.** A perfectly unlearned model (b) should show similar MIA metrics distribution for $\mathcal{D}_{\text{forget}}$ and $\mathcal{D}_{\text{holdout}}$. Unlearning methods may fail by under-unlearning $\mathcal{D}_{\text{forget}}$, making it similar to $\mathcal{D}_{\text{retain}}$ (c), or over-unlearning it, causing divergence from $\mathcal{D}_{\text{holdout}}$ (d).

**C2. No knowledge memorization**  When a model has unlearned a medical record, it should no longer be able to answer questions about that record. We measure a model $f$'s memorization of knowledge from the forget set $\mathcal{D}_{\text{forget}}$ as follows: for each example $x \in \mathcal{D}_{\text{forget}}$ associated with a question-answer pair $(q,a)$,[1] we gather the model's answer to the question $q$, denoted $f(q)$. We then average the ROUGE scores for all question-answer pairs in $\mathcal{D}_{\text{forget}}$ to compute the knowledge memorization score KnowMem:

$$\text{KnowMem}(f, \mathcal{D}_{\text{forget}}) := \frac{1}{|\mathcal{D}_{\text{forget}}|} \sum_{(q,a) \in \mathcal{D}_{\text{forget}}} \text{ROUGE}(f(q), a)$$

**C3. No privacy leakage**  As discussed previously, it is desirable that the unlearned model does not leak membership information indicating that $\mathcal{D}_{\text{forget}}$ was part of $\mathcal{D}_{\text{train}}$. To determine if a given example was used during training, *membership inference attack* (MIA) exploits distributional differences in certain statistics (e.g., loss) between training (member) and non-training (non-member) data: if the loss on the example is low, then it was likely used for training. Using MIAs to evaluate unlearning processes is a well-established practice as shown by prior research (Hayes et al., 2024; Triantafillou et al., 2023). An effective unlearning algorithm should eliminate such influence to reduce the attack's success rate. As shown in Figure 2, unlearning typically increases the loss on the example, but there are two possible ways that unlearning can fail to prevent privacy leakage: (1) *under-unlearning*, when the loss is not made large enough; and (2) *over-unlearning*, when the loss is made abnormally large. To accurately measure the privacy leakage, we employ Min-K% Prob (Shi et al., 2024a) , a state-of-the-art MIA method for LMs based on the loss, and compute the standard AUC-ROC score (Murakonda et al., 2021; Ye et al., 2022a) of discriminating $\mathcal{D}_{\text{forget}}$ (members) and $\mathcal{D}_{\text{holdout}}$ (non-members).[2] By comparing the AUC score with that of the retrained model, we define[3]

$$\text{PrivLeak} := \frac{\text{AUC}(f_{\text{unlearn}}; \mathcal{D}_{\text{forget}}, \mathcal{D}_{\text{holdout}}) - \text{AUC}(f_{\text{retrain}}; \mathcal{D}_{\text{forget}}, \mathcal{D}_{\text{holdout}})}{\text{AUC}(f_{\text{retrain}}; \mathcal{D}_{\text{forget}}, \mathcal{D}_{\text{holdout}})},$$

The PrivLeak metric for a good unlearning algorithm should be close to zero, whereas an over/under-unlearning algorithm will get a large positive/negative metric. More details about privacy leakage are discussed in Appendix B.1.

**Deployer expectations.**  Model deployers have their own considerations for using unlearning algorithms in the real world. Unlearning specific datapoints can unpredictably degrade model capabilities in ways that are difficult to recover. Moreover, deployers are expected to effectively accommodate somewhat large-scale forget sets and successive unlearning requests from data owners. As such, we consider three key metrics: (C4) **utility preservation** on the retain set, (C5) **scalability** to handle large-scale content removal, and (C6) **sustainability** to maintain performance over sequential unlearning requests.

**C4. Utility preservation.**  Model capabilities are often hard-won through expensive training procedures, so deployers would want an unlearning algorithm that preserves performance on the retain

---

[1]Examples of question-answer pairs derived from the original corpus can be found in Table 7.

[2]An MIA algorithm compares its score to a given threshold to classify a given datapoint as a member or non-member. The AUC-ROC is a single value that summarizes the overall performance of the MIA algorithm by measuring its ability to discriminate between members and non-members across all possible thresholds.

[3]Generally, $\text{AUC}(f_{\text{retrain}}; \mathcal{D}_{\text{forget}}, \mathcal{D}_{\text{holdout}}) \approx 0.5$, though sometimes there are intrinsic distribution shifts between $\mathcal{D}_{\text{forget}}$ and $\mathcal{D}_{\text{holdout}}$ that may bias the baseline away from 0.5.

Table 2: Examples of **MUSE**. Each corpus has `Verbatim` text and `Knowledge` sets (QA pairs derived from the original text) for evaluating verbatim and knowledge memorization. In NEWS, $\mathcal{D}_{\text{forget}}$ and $\mathcal{D}_{\text{retain}}$ are two disjoint sets of news articles. In BOOKS, $\mathcal{D}_{\text{forget}}$ is the Harry Potter book series while $\mathcal{D}_{\text{retain}}$ consists of wiki articles about the series. The sizes of the forget and retain sets are reported in tokens in (). Note that only the `Verbatim` texts within the Forget Set are included in our training data, while all `Knowledge` sets (QA pairs) serve for evaluations.

| Corpus | Forget Set | Retain Set |
|---|---|---|
| NEWS | **NEWS ARTICLE** (0.8 M tokens) | **NEWS ARTICLE** (1.6 M tokens) |
| | `MP Stuart McDonald has been appointed as the SNP's new treasurer` | `A father whose 12-year-old son was killed by an IRA bomb 30 years ago` |
| | **Q**: What position has Stuart McDonald MP been appointed to? **A**: The SNP's new treasurer | **Q**: Who was affected by the IRA bomb 30 years ago? **A**: A father whose 12-year-old son |
| BOOKS | **HARRY POTTER BOOKS** (1.1 M tokens) | **HARRY POTTER FANWIKI** (0.5 M tokens) |
| | `"There's more in the frying pan," said Aunt Petunia, turning eyes on her massive son.` | `This page contains a list of spells: Portuguese for 'open'.` |
| | **Q**: What does Aunt Petunia tell her son? **A**: There's more in the frying pan. | **Q**: What is the spell used to open things? **A**: Portuguese |

set. To quantify this, we evaluate the unlearned model's performance on the retain set using the knowledge memorization metric $\mathsf{KnowMem}(f_{\text{unlearn}}, \mathcal{D}_{\text{retain}})$.

**C5. Scalability.** We assess the scalability of unlearning methods by examining their performance on forget sets of varying sizes. Let $\mathcal{D}_u^c$ denote a forget set of size $c$, and $f_u^c$ be the corresponding unlearned model. For any data owner-valued metric such as utility preservation, we measure scalability by analyzing the trend of this metric as $c$ increases from small to large values.

**C6. Sustainability.** Machine unlearning operations often need to be applied sequentially, as data removal requests may arrive at different times.[4] We denote the unlearned model after processing the $k$-th request as $f_{u,k}$. To measure sustainability, we analyze the trend of any data owner-valued metric as the number of sequential unlearning requests $k$ increases.

## 3.2 EVALUATION CORPUS

**MUSE** considers two representative types of textual data that may frequently involve unlearning requests: news articles (*Tremblay v. OpenAI, Inc.*, 2023) and books (Eldan & Russinovich, 2023). These datasets are detailed as follows:

- **NEWS** consists of BBC news articles (Li et al., 2023b) collected after August 2023. All articles are randomly divided into (disjoint) forget, retain, and holdout sets.
- **BOOKS** consists of the Harry Potter book series. To simulate a real-world setting for testing utility preservation (**C4**), we include different types of materials in the forget and retain sets. The forget set contains the original books, while the retain set contains related content from the Harry Potter FanWiki,[5] representing domain knowledge that should be retained after unlearning.

For each corpus, we construct: 1) `Verbatim` text: the original text to assess the unlearning methods to remove verbatim memorization (**C1**), and 2) `Knowledge` set: a set of derived (question, answer) pairs based on the original texts to evaluate the unlearning method's effectiveness in purging learned knowledge and preventing knowledge memorization (**C2**). To create the Knowledge set, we partition the Verbatim text into excerpts and use GPT-4 (OpenAI, 2023) to generate (question, answer) pairs for each excerpt. When constructing the dataset, we perform deduplication between $\mathcal{D}_{\text{forget}}$ and $\mathcal{D}_{\text{retain}}$ by removing documents with over 70% similarity based on 3-grams. For more details about the dataset generation pipeline, see Appendix D.

Table 7 provides examples from the news and books corpora. The details of the dataset splits and dataset sizes are provided in Appendix D.

## 4 UNLEARNING METHODS

We evaluate eight efficient approximate unlearning methods belonging to four families of algorithms.

---

[4] For example, under GDPR, if Alice requests the removal of her data and Bob submits another removal request 31 days later, both requests must be fulfilled within 30 days. This requires the model deployer to first unlearn Alice's data and then process Bob's request on the updated model.

[5] harrypotter.fandom.com/wiki

**Four families of unlearning methods.** We first introduce four families of unlearning methods, which serve as the basis for the eight methods we evaluate.

- **Gradient Ascent** (GA) minimizes the likelihood of correct predictions on $\mathcal{D}_{\text{forget}}$ by performing gradient ascent on the cross-entropy loss (the opposite of conventional learning with gradient descent). GA has achieved mixed results: while Jang et al. (2023) found it effective for unlearning examples from the Enron email dataset (Klimt & Yang, 2004) with minimal performance degradation, Ilharco et al. (2023) reported that GA significantly harms general model utility when unlearning a high-toxicity subset of the Civil Comments dataset (Borkan et al., 2019).
- **Negative Preference Optimization** (NPO; Zhang et al., 2024b) treats the forget set as negative preference data and adapts the offline DPO objective (Rafailov et al., 2023) to tune the model to assign low likelihood to the forget set without straying too far from the original model $f_{\text{target}}$.

$$\mathcal{L}_{\text{NPO}}(\theta) = -\frac{2}{\beta} \mathbb{E}_{x \sim \mathcal{D}_{\text{forget}}} \left[ \log \sigma \left( -\beta \log \frac{f_\theta(x)}{f_{\text{target}}(x)} \right) \right],$$

where $f_\theta$ refers to the model that undergoes unlearning, $\sigma$ is the sigmoid function, and $\beta$ is a hyperparameter that controls the allowed divergence of $f_\theta$ from its initialization $f_{\text{target}}$. Following Rafailov et al. (2023); Zhang et al. (2024b), we fix $\beta = 0.1$ in our experiments.

- **Task Vectors** (Ilharco et al., 2023) derived from straightforward arithmetic on the model weights can effectively steer neural network behavior. We adapt task vectors to perform unlearning in two stages. First, we train $f_{\text{target}}$ on $\mathcal{D}_{\text{forget}}$ until the model overfits, yielding a reinforced model $f_{\text{reinforce}}$. We then obtain a task vector related to $\mathcal{D}_{\text{forget}}$ by calculating the weight difference between $f_{\text{target}}$ and $f_{\text{reinforce}}$. To achieve unlearning, we subtract this task vector from $f_{\text{target}}$'s weights, intuitively moving the model away from the direction it used to adapt to $\mathcal{D}_{\text{forget}}$ − i.e., $f_{\text{unlearn}} = f_{\text{target}} - (f_{\text{reinforce}} - f_{\text{target}})$.
- **Who's Harry Potter** (WHP; Eldan & Russinovich, 2023) defines the unlearned model $f_{\text{unlearn}}$ as the interpolation between the target model $f_{\text{target}}$ and the reinforced model $f_{\text{reinforce}}$. Let $p_f(\cdot|x)$ denote the token distribution parametrized by the model $f$ when given a prompt $x$ as input. Then, concretely, for any input $x$, WHP samples the next token from

$$p_{f_{\text{unlearn}}}(\cdot|x) = p_{f_{\text{target}}}(\cdot|x) - \alpha(p_{f_{\text{reinforce}}}(\cdot|x) - p_{f_{\text{target}}}(\cdot|x))$$

where $\alpha$ is a hyperparameter that controls the interpolation between the two models.

**Two regularizers for utility preservation.** GA and NPO are not explicitly designed for utility preservation, so we discuss several regularization strategies that either improve the performance on the retain set or ensure the unlearned model remains close to the target model during unlearning.

- **Gradient Descent on the Retain Set** (GDR; Liu et al., 2022; Maini et al., 2024; Zhang et al., 2024b) augments the unlearning objective with a standard gradient descent learning objective on the cross-entropy of the retain set $\mathcal{D}_{\text{retain}}$ to more directly train the model to maintain its performance on $\mathcal{D}_{\text{retain}}$.
- **KL Divergence Minimization on the Retain Set** (KLR; Maini et al., 2024; Zhang et al., 2024b) encourages the unlearned model's probability distribution $p_{f_{\text{unlearn}}}(\cdot|x)$ to be close to the target model's distribution $p_{f_{\text{target}}}(\cdot|x)$ on inputs from the retain set $x \in \mathcal{D}_{\text{retain}}$.

**List of methods.** We combine GA and NPO with the two regularizers GDR and KLR,[6] which yields four new combinations. Hence, we end up with a total of 8 candidate unlearning methods: GA, $\text{GA}_{\text{GDR}}$, $\text{GA}_{\text{KLR}}$, NPO, $\text{NPO}_{\text{GDR}}$, $\text{NPO}_{\text{KLR}}$, Task Vector, and WHP. In general, the cost of the approximate unlearning method is negligible compared to retraining. Note that the methods with regularizers ($\text{GA}_{\text{GDR}}$, $\text{GA}_{\text{KLR}}$, $\text{NPO}_{\text{GDR}}$, $\text{NPO}_{\text{KLR}}$) require access to the distribution of $\mathcal{D}_{\text{retain}}$). Details about the efficiency of these methods are reported in Appendix B.4.

## 5 EXPERIMENTS

We evaluate the eight representative unlearning methods using the experimental setup described in §5.1. We present the results for data owner expectations in §5.2 and for deployer expectations in §5.3.

---

[6]These regularizers are not compatible with Task Vector and WHP, because Task Vector involves purposefully overfitting a model to $\mathcal{D}_{\text{forget}}$ when deriving the task vector, and WHP is a test-time technique where the unlearning operation involves no optimization by itself.

Table 3: **Most unlearning methods effectively remove verbatim and knowledge memorization but significantly impact utility and privacy.** We evaluate the 8 algorithms described in §4 on 4 of the criteria in **MUSE**. We include the results of $f_{\text{retrain}}$ for reference and calculate the relative ratio compared to the reference model. We highlight the ratio in blue if the unlearning algorithm satisfies the criterion and highlight it in orange otherwise. We define privacy leakage as negligible when it falls within the range of -5% to +5%. Large positive values suggest over-unlearning, while large negative values suggest under-unlearning (see §3.1). This table covers the results for C1 to C4, while results for C5 and C6 are shown in Figure 6.

| | **C1. No Verbatim Mem.** VerbMem on $\mathcal{D}_{\text{forget}}$ (↓) | | **C2. No Knowledge Mem.** KnowMem on $\mathcal{D}_{\text{forget}}$ (↓) | | **C3. No Privacy Leak.** PrivLeak ($\in [−5\%, 5\%]$) | | **C4. Utiltiy Preserv.** KnowMem on $\mathcal{D}_{\text{retain}}$ (↑) | |
|---|---|---|---|---|---|---|---|---|
| | | | | **NEWS** | | | | |
| Target $f_{\text{target}}$ | 58.4 | | 63.9 | | −99.8 | | 55.2 | |
| Retrain $f_{\text{retrain}}$ | **20.8** | | **33.1** | | **0.0** | | **55.0** | |
| GA | 0.0 | ↓100% | 0.0 | ↓100% | 5.2 | over-unlearn | 0.0 | ↓100% |
| GA$_{\text{GDR}}$ | 4.9 | ↓76.5% | 31.0 | ↓6.3% | 108.1 | over-unlearn | 27.3 | ↓50.3% |
| GA$_{\text{KLR}}$ | 27.4 | ↑31.4% | 50.2 | ↑51.5% | −96.1 | under-unlearn | 44.8 | ↓18.5% |
| NPO | 0.0 | ↓100% | 0.0 | ↓100% | 24.4 | over-unlearn | 0.0 | ↓100.0% |
| NPO$_{\text{GDR}}$ | 1.2 | ↓94.4% | 54.6 | ↑64.8% | 105.8 | over-unlearn | 40.5 | ↓26.3% |
| NPO$_{\text{KLR}}$ | 26.9 | ↑29.0% | 49.0 | ↑48.1% | −95.8 | under-unlearn | 45.4 | ↓17.4% |
| Task Vector | 57.2 | ↑174.7% | 66.2 | ↑100.0% | −99.8 | under-unlearn | 55.8 | ↑1.5% |
| WHP | 19.7 | ↓5.6% | 21.2 | ↓35.9% | 109.6 | under-unlearn | 28.3 | ↓48.5% |
| | | | | **BOOKS** | | | | |
| Target $f_{\text{target}}$ | 99.8 | | 59.4 | | −57.5 | | 66.9 | |
| Retrain $f_{\text{retrain}}$ | **14.3** | | **28.9** | | **0.0** | | **74.5** | |
| GA | 0.0 | ↓100% | 0.0 | ↓100% | −25.0 | under-unlearn | 0.0 | ↓100% |
| GA$_{\text{GDR}}$ | 0.0 | ↓100% | 0.0 | ↓100% | −26.5 | under-unlearn | 10.7 | ↓85.6% |
| GA$_{\text{KLR}}$ | 16.0 | ↑11.4% | 21.9 | ↓24.4% | −40.2 | under-unlearn | 37.2 | ↓50.0% |
| NPO | 0.0 | ↓100% | 0.0 | ↓100% | −24.3 | under-unlearn | 0.0 | ↓100% |
| NPO$_{\text{GDR}}$ | 0.0 | ↓100% | 0.0 | ↓100% | −30.8 | under-unlearn | 22.8 | ↓69.4% |
| NPO$_{\text{KLR}}$ | 17.0 | ↑18.2% | 25.0 | ↓13.4% | −43.5 | under-unlearn | 44.6 | ↓40.1% |
| Task Vector | 99.7 | ↑595.0% | 52.4 | ↑81.2% | −57.5 | under-unlearn | 64.7 | ↓13.1% |
| WHP | 18.0 | ↑25.2% | 55.7 | ↑92.9% | 56.5 | over-unlearn | 63.6 | ↓14.6% |

## 5.1 EXPERIMENTAL SETUP

**Retrained and target models.** We start with a general pretrained base model $f_0$, and finetune two models: $f_{\text{target}}$ on $\mathcal{D}_{\text{forget}} \cup \mathcal{D}_{\text{retain}}$, and $f_{\text{retrain}}$ on $\mathcal{D}_{\text{retain}}$ only. See Appendix B.3 for details about finetuning. For each unlearning algorithm $\mathcal{U}$, we further generate the unlearned model $f_{\text{unlearn}} = \mathcal{U}(f_{\text{target}}, \mathcal{D}_{\text{forget}}, \mathcal{D}_{\text{retain}})$. We ensure that $f_0$ has no access to $\mathcal{D}_{\text{forget}}, \mathcal{D}_{\text{retain}}, \mathcal{D}_{\text{holdout}}$. Therefore, for NEWS, we use $f_0 = $ LLaMA-2 7B (Touvron et al., 2023), which was released *before* the BBC news articles we use to construct our benchmarks; and for BOOKS, we use $f_0 = $ ICLM-7B (Shi et al., 2024b), which does *not* contain the Harry Potter books in its pretraining data.

**Unlearning experimental configuration.** Following prior work (Maini et al., 2024), we run GA, NPO, and their regularized variants using the AdamW optimizer (Loshchilov & Hutter, 2017) with a constant learning rate of $10^{-5}$ and a batch size of 32. We employ the stopping criteria as follows: if the utility (i.e., KnowMem on $\mathcal{D}_{\text{retain}}$) of a model undergoing unlearning drops below that of $f_{\text{retrain}}$ within 10 epochs of unlearning, we stop at the first epoch where this condition holds; otherwise, we take a checkpoint from the 10th epoch. For Task Vector and WHP, to obtain the reinforced model for unlearning, we fine-tune the target model for 10 epochs using the same learning rate and batch size. Further details on the model fine-tuning and unlearning can be found in Appendix B.3.

## 5.2 RESULTS: DATA OWNER EXPECTATIONS

We first analyze how eight unlearning methods meet data owner expectations (C1, C2 & C3 in §3.1).

**C1&C2. Most methods are effective for unlearning memorization.** As shown in Table 3, most unlearning methods perform exceptionally well in [C1. No verbatim memorization] and [C2. No knowledge memorization], often reducing VerbMem and KnowMem even beyond the levels achieved by the retrained model. Notably, some methods, such as GA and NPO, achieve a score of 0 for both VerbMem and KnowMem, meaning that these methods completely prevent the unlearned models from producing any text related to the forget set. However, as we will see later, these reductions often come at the cost of significant utility loss on the retain set.

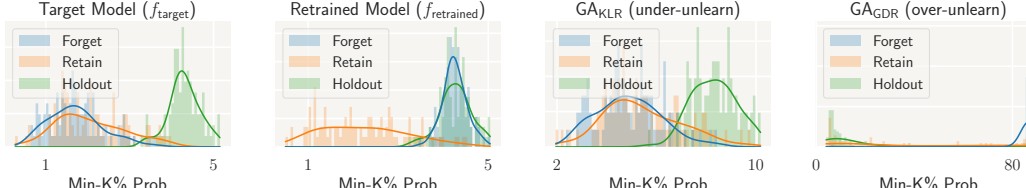

Figure 3: **Distribution of Min-K% Prob, an MIA metric, for $\mathcal{D}_{\text{forget}}$, $\mathcal{D}_{\text{holdout}}$, and $\mathcal{D}_{\text{retain}}$.** Consistent with the expected pattern in Figure 2, $f_{\text{retrain}}$ shows perfect unlearning, with the overlapping distributions for $\mathcal{D}_{\text{forget}}$ and $\mathcal{D}_{\text{holdout}}$. Existing approximate unlearning methods typically either under-unlearn or over-unlearn. For example, $\text{GA}_{\text{KLR}}$ shows slight under-unlearning, while $\text{GA}_{\text{GDR}}$ over-unlearns, pushing the Min-K% Prob of $\mathcal{D}_{\text{forget}}$ to an extreme level.

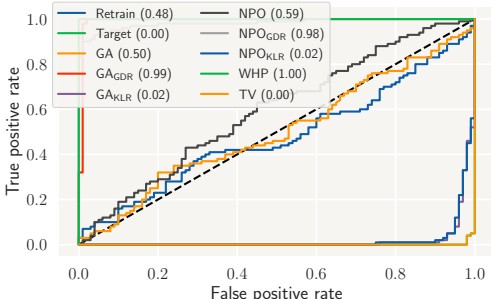

Figure 4: **ROC curves for $\mathcal{D}_{\text{forget}}$ vs. $\mathcal{D}_{\text{holdout}}$ on NEWS using Min-K% Prob, with AUC scores in parentheses.** AUC≈0.5 (i.e., $f_{\text{retrain}}$) means no significant distribution difference between two sets (i.e., no membership leakage). Most unlearning methods show under-unlearn (AUC≪0.5) or over-unlearn (AUC ≫0.5).

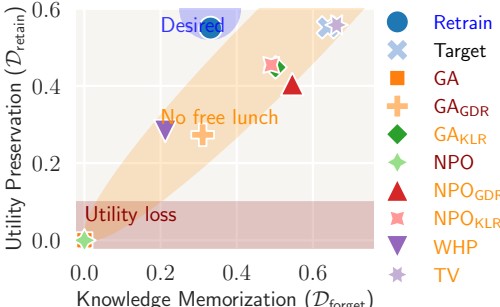

Figure 5: **Utility preservation vs. knowledge memorization on BBC.** $f_{\text{retrain}}$ maintains high utility on $\mathcal{D}_{\text{retain}}$ while showing low knowledge memorization on $\mathcal{D}_{\text{forget}}$. GA and NPO without regularizers show significant utility loss, collapsing to the origin. Every other unlearning method unlearns the knowledge on $\mathcal{D}_{\text{forget}}$ at the cost of utility.

**C3. Unlearning leads to privacy leakage.** Most unlearning methods reveal the membership of $\mathcal{D}_{\text{forget}}$ in $\mathcal{D}_{\text{train}}$ through under-unlearning (PrivLeak ≪ 0) or over-unlearning (PrivLeak ≫ 0), as shown in Table 3. We further examine the effectiveness of membership inference by plotting ROC curves in Figure 4. The deviation from the diagonal line indicates the attacker's advantage over random guessing. We observe that the Min-K% Prob based attack achieves AUC ≈ 0 on $f_{\text{target}}$, confirming its effectiveness. Meanwhile, the ROC curve for $f_{\text{retrain}}$ closely follows the diagonal line (AUC = 0.47), suggesting that perfect unlearning ensures MIA is no more effective than random guessing. Among the approximate unlearning methods, GA and $\text{NPO}_{\text{GDR}}$ without regularizers consistently over-unlearn (AUC > 0.7), whereas KLR-regularized methods ($\text{NPO}_{\text{KLR}}$ and $\text{GA}_{\text{KLR}}$) tend to under-unlearn and barely improve privacy leakage over $f_{\text{target}}$. WHP also deviates from the diagonal significantly.

In Figure 3, we further visualize the distribution of Min-K% Prob, the MIA metric computed across $\mathcal{D}_{\text{forget}}$, $\mathcal{D}_{\text{retain}}$, and $\mathcal{D}_{\text{holdout}}$. The behavior of $f_{\text{target}}$ and $f_{\text{retrain}}$ mirrors the patterns sketched in Figure 2, where $\mathcal{D}_{\text{forget}}$ and $\mathcal{D}_{\text{retain}}$ are distinguishable in $f_{\text{target}}$ but overlap in $f_{\text{retrain}}$. Existing approximate unlearning methods typically either under-unlearn or over-unlearn. For example, $\text{GA}_{\text{KLR}}$ does not sufficiently increase the Min-K% Prob metric for $\mathcal{D}_{\text{forget}}$ to align with the distribution of $\mathcal{D}_{\text{holdout}}$, indicating under-unlearning. On the other hand, $\text{NPO}_{\text{GDR}}$ over-unlearns, significantly raising the MIA metric across all datasets and especially for $\mathcal{D}_{\text{forget}}$.

## 5.3 RESULTS: DEPLOYMENT CONSIDERATIONS

**C4. Unlearning significantly degrades model utility.** Table 3 [C4 Utility Preserv.] shows that all unlearning methods compromise the model's utility by $24.2\% \sim 100\%$. Notably, several methods (GA, $\text{GA}_{\text{GDR}}$, $\text{NPO}_{\text{GDR}}$) lead to complete utility loss, rendering the unlearned models practically unusable. Figure 5 illustrates the trade-offs between utility preservation on $\mathcal{D}_{\text{retain}}$ and knowledge memorization on $\mathcal{D}_{\text{forget}}$. An ideal unlearned model should mimic the behavior of $f_{\text{retrain}}$ (desired region) by achieving a low level of memorization on $\mathcal{D}_{\text{forget}}$ while maintaining its utility. However, most methods, such as $\text{GA}_{\text{KLR}}$, $\text{NPO}_{\text{KLR}}$, and WHP, unlearn the knowledge on $\mathcal{D}_U$ at the cost of utility.

**C5. Unlearning methods scale poorly with forget set sizes.** To evaluate the robustness of the unlearning methods to larger forget sets, we collect additional news articles from the same distribution to scale our NEWS corpus from 0.8M tokens to 3.3M tokens and observe the utility preservation at four different forget set sizes. As shown in Figure 6 (a), the model utility decrease with the size of the forget set and achieves a minimum at the largest size.

**C6. Unlearning methods cannot sustainably accommodate sequential unlearning requests.** To evaluate the robustness of these unlearning methods to more than one unlearning requests, we sequentially apply $k$ unlearning processes, each with respect to a different forget set. To simulate sequential unlearning, we partition the extended NEWS forget set (comprised of 3.3M tokens) into four disjoint folds (each containing 0.8M tokens) and apply the unlearning methods to each fold in a sequential manner.

We again select utility preservation as the target metric for comparison. As shown in Figure 6 (b), the performance of an unlearned model tends

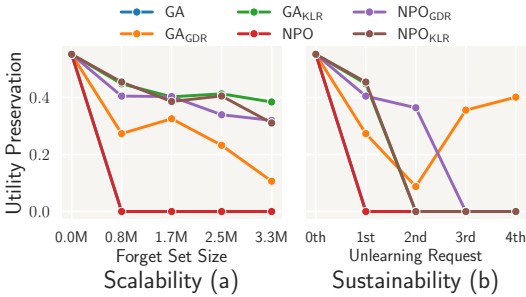

Figure 6: **The performance of GA, NPO, and their regularized variants, measured by utility preservation, degrades with larger forget set sizes (a) and sequential unlearning requests (b).**

to decrease significantly with respect to the number of unlearning requests, indicating that current unlearning methods are not yet ready to handle sequential unlearning in a sustainable manner.

## 6 RELATED WORK

**Machine unlearning for non-language model applications.** Machine unlearning is a long-running, well-studied topic. Several studies have explored exact unlearning, aiming to make the unlearned model ($f_{\text{unlearn}}$) exactly identical to the reference model ($f_{\text{retrain}}$). As expected, this can only be accomplished in simple models like SVMs (Cauwenberghs & Poggio, 2000; Tveit et al., 2003; Romero et al., 2007; Karasuyama & Takeuchi, 2010) or naive Bayes models (Cao & Yang, 2015). Another approach is to ensure that the unlearned model $f_{\text{unlearn}}$ is probabilistically indistinguishable from $f_{\text{retrain}}$ (Ginart et al., 2019; Guo et al., 2020), and this view of certifiable unlearning is closely related to differential privacy (Dwork et al., 2006b;a). This rigorous definition of unlearning has inspired several theoretical works that characterize the feasibility of unlearning in convex and non-convex models, but those proposed algorithms are too computationally costly to operate on modern-day LMs (Izzo et al., 2021; Neel et al., 2021; Ullah et al., 2021; Sekhari et al., 2021; Gupta et al., 2021). Several more tractable unlearning algorithms have been proposed (Borkan et al., 2019; Ginart et al., 2019; Thudi et al., 2022; Chourasia & Shah, 2023) with broader applications such as image classification (Ginart et al., 2019; Golatkar et al., 2020a), text-to-image generation (Gandikota et al., 2023; Zhang et al., 2023; Fan et al., 2023), Federated Learning (Liu et al., 2020; Che et al., 2023; Halimi et al., 2022; Huang et al., 2022) and Recommender Systems (Li et al., 2024b).

**Machine unlearning for language models: methods and applications.** Machine unlearning has recently found its way into language model applications. In §4, we discuss some standard unlearning methods based on parameter optimization, like the Gradient Ascent and its variance. Other notable non-training-based unlearning methods include localization-informed unlearning (Meng et al., 2022; Wu et al., 2023; Wei et al., 2024a), which involves identifying model units (e.g., layers, neurons) closely related to the unlearning data or tasks and then locally editing and modifying the units. In-context unlearning (Pawelczyk et al., 2023) offers another approach, treating the model as a black box and modifying its output results using external knowledge.

Machine unlearning has also been applied to various downstream language model tasks, though the unit of machine unlearning may differ from what we study in this work. Our evaluation focuses on unlearning specific examples or datasets, aiming to make LMs forget either the phrasing or the content knowledge of targeted data, while preserving their utility for data not targeted for removal. This is crucial for ensuring privacy and copyright compliance. In addition to this specific unlearning, there's also a broader application similar to model editing, where outdated information is replaced with new knowledge (Pawelczyk et al., 2023; Yu et al., 2023; Belrose et al., 2024). Moreover, efforts have been made to eliminate harmful behaviors in language models by creating toxicity benchmarks and enhancing safety measures (Lu et al., 2022; Yao et al., 2023; Li et al., 2024a; Zhang et al., 2024b).

Despite these varied approaches to unlearning at different operational and knowledge levels, the evaluation principles we propose such as preserving utility, ensuring scalability, and maintaining sustainability—are relevant across these contexts.

**Machine unlearning for language models: evaluation.** Evaluating machine unlearning methods for language model applications is also critical. Most previous studies have focused this evaluation on specific tasks such as question answering or sentence completion. For example, Eldan & Russinovich (2023) experiment with unlearning to forget Harry Potter books and demonstrate the effectiveness of their methods by showing that familiarity scores, measured through completion-based, token-probability-based, and question-answering evaluations, significantly decline post-unlearning. Lynch et al. (2024) further suggest comparing unlearned models with perfectly retrained models. Their evaluation finds that while familiarity scores with the forget set may drop post-unlearning, they still remain higher than those of the retrained model. Wei et al. (2024b) evaluate the feasibility of using unlearning techniques to prevent language models from generating copyrighted content. The closest work to ours is TOFU (Maini et al., 2024), a benchmark featuring 200 synthetic author profiles, each with 20 question-answer pairs, divided into forget and retain sets. However, TOFU is relatively small-scale (0.15M tokens) and focuses on the evaluation of question answering. Additionally, current evaluations focus on limited aspects of data owner expectations and do not adequately reflect real-world deployment considerations, such as scalability and potential sequential unlearning requests. In contrast, **MUSE** formally defines different unlearning scopes and corresponding metrics, resulting in a systematic six-way evaluation featuring both data owners' and deployers' expectations. The evaluation uses a large-scale corpus of over 6 million tokens, separated into verbatim text and knowledge sets. We also note that some of our findings align with previous evaluations. For example, our observation that over- or under-unlearn can exacerbate privacy leakage (§5.2) is consistent with the recent work by Hayes et al. (2024). Our findings align with the the concurrent study by Shumailov et al. (2024) showing that unlearning gives a false sense of security.

**Survey papers.** We direct readers to several insightful survey papers for further reading. For non-LLM applications, notable surveys include Shintre et al. (2019); Nguyen et al. (2022); Thudi et al. (2022); Xu et al. (2023). Additionally, the NeurIPS 2023 machine unlearning competition for image classification[7] is a valuable source of empirical methods tailored for this specific application (Triantafillou et al., 2023). For language model applications, Si et al. (2023) categorize unlearning methods into different families and summarize datasets for evaluating unlearning. Liu et al. (2024) review LM unlearning algorithms by targets and methods, discuss the effectiveness and efficiency of existing approaches and emphasize the importance of clearly defining the unlearning scope.

## 7 CONCLUSION

In this work, we propose **MUSE**, a comprehensive machine unlearning evaluation benchmark that highlights six desirable properties from the perspectives of both data owners and model deployers. We find that current unlearning methods successfully prevent the model's memorization of content at a significant cost to utility on data not intended for removal. They also lead to severe privacy leakage and cannot sustainably accommodate successive unlearning requests or large-scale content removal. These findings highlight the need for future research into more robust unlearning methods.

**Limitations.** While **MUSE** provides a systematic benchmark for evaluating unlearning algorithms, it does not consider all possible considerations. For example, data owners may have additional expectations, such as ensuring their information cannot be probed from intermediate activations (Song & Raghunathan, 2020) or receiving formal guarantees of unlearning success (Sekhari et al., 2021; Gupta et al., 2021; Ghazi et al., 2023). Similarly, deployers may expect other capabilities, like fine-tuning and in-context learning, to be preserved, and may prefer unlearning algorithms that are both computationally efficient and storage-wise cheap (e.g. does not need to keep a copy of the retain set). **MUSE** currently evaluates unlearning for language models using books and news articles, but it could be extended to other corpora, such as medical notes (Johnson et al., 2016; 2020) and emails (Klimt & Yang, 2004), which often involve privacy concerns (Li et al., 2023a; Huang et al., 2023). We also plan to evaluate different-sized LMs in the future. Finally, our approach can be generalized to construct multi-faceted benchmarks for multimodal models (Golatkar et al., 2020b; Cheng & Amiri, 2023; Zhang et al., 2024c). Further discussion on broader impact are in Appendix A.

---

[7]https://unlearning-challenge.github.io

## REPRODUCIBILITY STATEMENT

We are committed to making all aspects of our work fully open-source, providing comprehensive instructions to guarantee reproducibility.

**Models** The weights for our original models, $f_{\text{target}}$ and $f_{\text{unlearn}}$, will be released under the Apache 2.0 open-source license.

**Data** Our benchmark datasets will be made available under open-source licenses.

**Code** We will provide the code for all baseline methods, evaluation scripts used for benchmarking, as well as the code for visualizations and analysis presented in this paper. Detailed instructions will accompany our code to ensure precise reproducibility.

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

# Appendices

# A    BROADER IMPACT

As LMs are deployed broadly and publicly, there is mounting legal and social pressure on deployers to release models that permit effective unlearning when requested by data owners (European Parliament & Council of the European Union; *DOE 1 v. GitHub, Inc.*, N.D. Cal. 2022; *Tremblay v. OpenAI, Inc.,*, 2023). These incentives have prompted a flurry of new unlearning algorithms stemming from different technical perspectives. As such, systematic evaluation of the strengths and weaknesses of these methods when executing realistic unlearning requests on popular models is essential. **MUSE** disentangles several desirable properties of unlearning algorithms and finds that no existing algorithm is able to satisfy all of the data owner and deployer considerations. We hope that our fine-grained, multi-faceted framework facilitates the improvement of unlearning algorithms. Moreover, we expect that the general approach of designing metrics to balance the considerations of various stakeholders is flexible and can adapt to the rapidly shifting legal, social, and economic landscape.

We also acknowledge the potential negative impacts of our study. One limitation of our evaluation benchmark is that we do not have comprehensive study of how unlearning would impact the model performance for different user bases, especially underrepresented groups. However, we note proper handling and evaluation of fairness issues in unlearning is still an active ongoing research area (Zhang et al., 2024a; Oesterling et al., 2024), therefore we leave it as future work. Additionally, our work may be misinterpreted towards skepticism regarding the broader use of machine unlearning, as our current evaluation reveals that existing unlearning methods are not yet ready for effective real-world deployment. However, machine unlearning, especially for large language models, is a young and active research area and new algorithms are constantly being proposed. We emphasize that our results is not a criticism of the paradigm of machine unlearning, but a study of the potential downsides of existing methods and a call for better algorithms. We believe our benchmark is an important step towards guiding future algorithm design of machine unlearning research towards more realistic deployment scenarios.

# B  EXPERIMENTAL DETAILS

## B.1  THREAT MODEL FOR PRIVACY LEAKAGE

We provide further clarification on the threat model considered for our C3: no privacy leakage. We assume an attacker with access to a trained model aims to determine whether a specific example (belonging to a particular data owner) was part of the training set. Prior work on membership inference attacks (MIA) demonstrates that these attacks can detect a training sample's influence on the trained model, and use that to distinguish between training and non-training samples. Therefore, an effective unlearning algorithm should eliminate such influence to reduce the attack's success rate, making the model unable to distinguish between a true non-training example and one that was trained and subsequently unlearned.

Note that in this threat model, we assume the attacker only have access to the final unlearned model, because if both the target model and the unlearned models are available at the same time, then there is no point to perform unlearning.

## B.2  COMPUTE CONFIGURATIONS

All experiments are conducted on 8 NVIDIA A40 GPU cards in a single node.

## B.3  EXPERIMENTAL SETUP

**Finetuning details.**  As described in §5.1, for NEWS, we start from $f_0 = $ LLaMA-2 7B (Touvron et al., 2023) and finetune the model on the BBC news articles for 5 epochs with a constant learning rate of $10^{-5}$ and a batch size of 32 . For BOOKS, we start from $f_0 = $ ICLM 7B (Touvron et al., 2023) and finetune the model on the Harry Potter books with same set of hyperparameters.

**Unlearning details.**  For all the unlearning methods in Table 3, we use a constant learning rate of $10^{-5}$ and a batch size of 32. For $f_{\text{reinforced}}$ used in WHP and Task Vector, we fine-tune $f_{\text{target}}$ for 10 epochs.

Before evaluation, for each unlearning method, we select its optimal epoch or $\alpha$ (both of which are parameters that control a degree of unlearning) by using our unlearning stopping criteria based on the unlearned model's utility on $\mathcal{D}_{\text{retain}}$ compared to that of $f_{\text{retrain}}$. The chosen epochs or $\alpha$'s for each method are listed below.

Table 4: Optimal epochs or $\alpha$'s for each unlearning method.

| Unlearning Method | NEWS | BOOKS |
|---|---|---|
| GA | epoch 1 | epoch 1 |
| GA$_{\text{GDR}}$ | epoch 7 | epoch 1 |
| GA$_{\text{KLR}}$ | epoch 10 | epoch 5 |
| NPO | epoch 1 | epoch 1 |
| NPO$_{\text{GDR}}$ | epoch10 | epoch 1 |
| NPO$_{\text{KLR}}$ | epoch 10 | epoch 4 |
| Task Vector | $\alpha = 2^9$ | $\alpha = 2^9$ |
| WHP | $\alpha = 2^2$ | $\alpha = 2^8$ |

## B.4  EFFICIENCY OF UNLEARNING METHODS

We report the efficiency of unlearning methods in Table 5, measured by the wall-clock time for a single gradient update step of unlearning. The time measurements were conducted using 8 NVIDIA A40 GPUs on a single node, with a batch size of 32 and an input length of 2048 tokens. Each step corresponds to one gradient update processing a total of 65,536 tokens ($32 \times 2048$ tokens). For Task Vector and WHP, each step represents one iteration of fine-tuning to create the reinforced model.

Table 5: Wall-clock time and total GPU hours required for each unlearning method.

| Unlearning Method | Time (Seconds/Step) | Total Time (GPU Hours) |
|---|---|---|
| Retrain | - | 184320 |
| GA | 4.14 | 0.56 |
| GA$_{\text{GDR}}$ | 6.05 | 0.82 |
| GA$_{\text{KLR}}$ | 7.58 | 1.03 |
| NPO | 5.68 | 0.77 |
| NPO$_{\text{GDR}}$ | 7.59 | 1.03 |
| NPO$_{\text{KLR}}$ | 9.11 | 1.24 |
| Task Vector | 4.14 | 1.12 |
| WHP | 4.14 | 1.12 |

# C  MORE EXPERIMENTAL RESULTS

## C.1  CONFIDENCE INTERVALS FOR C1, C2 AND C4 IN TABLE 3

We compute confidence intervals for C1, C2, and C4 (Mean ROUGE-L F1) using bootstrapping[8]. For each mean ROUGE-L score reported in Table 3, we draw 9,999 bootstrap resamples and calculate a two-tailed 95% confidence interval using the "percentage" method.

Table 6: 95% confidence intervals computed for mean Rouge-L scores used in C1, C2, and C4.

| | C1. No Verbatim Mem. VerbMem on $\mathcal{D}_{\text{forget}}$ ($\downarrow$) | | C2. No Knowledge Mem. KnowMem on $\mathcal{D}_{\text{forget}}$ ($\downarrow$) | | C4. Utiltiy Preserv. KnowMem on $\mathcal{D}_{\text{retain}}$ ($\uparrow$) | |
|---|---|---|---|---|---|---|
| **NEWS** | | | | | | |
| Target $f_{\text{target}}$ | 58.4 | [54.1, 62.9] | 63.9 | [58.7, 69.0] | 55.2 | [50.7, 59.9] |
| Retrain $f_{\text{retrain}}$ | **20.8** | [18.5, 23.7] | **33.1** | [26.8, 39.5] | **55.0** | [50.3, 59.8] |
| GA | 0.0 | [0.0, 0.0] | 0.0 | [0.0, 0.0] | 0.0 | [0.0, 0.0] |
| GA$_{\text{GDR}}$ | 4.9 | [4.5, 5.2] | 31.0 | [24.2, 38.0] | 27.3 | [21.9, 33.0] |
| GA$_{\text{KLR}}$ | 27.4 | [25.1, 29.9] | 50.2 | [43.1, 56.9] | 44.8 | [39.2, 50.5] |
| NPO | 0.0 | [0.0, 0.0] | 0.0 | [0.0, 0.0] | 0.0 | [0.0, 0.0] |
| NPO$_{\text{GDR}}$ | 1.2 | [0.3, 2.3] | 54.6 | [47.5, 61.5] | 40.5 | [34.7, 46.2] |
| NPO$_{\text{KLR}}$ | 26.9 | [24.7, 29.3] | 49.0 | [41.8, 61.5] | 45.4 | [39.8, 51.1] |
| Task Vector | 57.2 | [52.6, 62.0] | 66.2 | [61.3, 71.2] | 55.8 | [51.0, 60.6] |
| WHP | 19.7 | [17.8, 21.6] | 21.2 | [16.0, 26.7] | 28.3 | [23.3, 33.4] |
| **BOOKS** | | | | | | |
| Target $f_{\text{target}}$ | 99.8 | [99.8, 99.9] | 59.4 | [52.7, 66.0] | 66.9 | [59.6, 73.8] |
| Retrain $f_{\text{retrain}}$ | **14.3** | [13.6, 15.1] | **28.9** | [22.1, 35.7] | **74.5** | [68.4, 80.0] |
| GA | 0.0 | [0.0, 0.0] | 0.0 | [0.0, 0.0] | 0.0 | [0.0, 0.0] |
| GA$_{\text{GDR}}$ | 0.0 | [0.0, 0.0] | 0.0 | [0.0, 0.0] | 10.7 | [6.2, 15.7] |
| GA$_{\text{KLR}}$ | 16.0 | [14.8, 17.2] | 21.9 | [16.4, 27.7] | 37.2 | [29.5, 45.0] |
| NPO | 0.0 | [0.0, 0.0] | 0.0 | [0.0, 0.0] | 0.0 | [0.0, 0.0] |
| NPO$_{\text{GDR}}$ | 0.0 | [0.0, 0.0] | 0.0 | [0.0, 0.0] | 22.8 | [16.1, 30.1] |
| NPO$_{\text{KLR}}$ | 17.0 | [15.7, 18.2] | 25.0 | [19.0, 31.5] | 44.6 | [36.5, 52.8] |
| Task Vector | 99.7 | [99.6, 99.8] | 52.4 | [45.0, 59.7] | 64.7 | [57.1, 71.8] |
| WHP | 18.0 | [16.4, 19.7] | 55.7 | [48.6, 62.8] | 63.6 | [56.3, 70.9] |

---

[8]https://docs.scipy.org/doc/scipy/reference/generated/scipy.stats.bootstrap.html

# D DATASET DETAILS

## D.1 GPT-GENERATED QA PAIRS

We begin the generation by partitioning the `Verbatim` text of each corpus into a set of 2048-token excerpts using LLaMA-2's tokenizer. For each QA pair to generate, we randomly sample an excerpt from this set and prompt GPT-4 (`gpt-4o-2024-05-13`) to create a JSON object with two fields: "*question*" (a question that can only be answered using specific information from the excerpt) and "*answer*" (an answer to the "question" extracted verbatim from the excerpt). We validate and exclude any pairs whose answers cannot be found verbatim in their corresponding excerpts. This verbatim requirement ensures that our `Knowledge` set is used precisely to evaluate the model's ability to correctly associate questions with relevant portions of the training data.

For each QA pair to generate, we initiate a new conversation with GPT-4 with its corresponding excerpt. The instruction begins with a system prompt that specifies the desired format of generated QA pairs as follows:

> **System Prompt for Generating QAs with GPT-4**
>
> You will be provided with an excerpt of text. Your goal is to create a question-answer pair that assesses reading comprehension and memorization, ensuring that the question can only be answered using details from the excerpt.
>
> Please submit your response in a JSON format with the following fields:
> - "question": A single question related to the excerpt. The question should be specific enough that it does not allow for an answer other than the one you provide. In particular, it should not be answerable based on common knowledge alone. Also, a few words extracted from the excerpt must suffice in answering this question.
> - "answer": A precise answer extracted verbatim, character-by-character from the excerpt. The answer to this question must be short, phrase-level at most. The length of the extraction should be minimal, providing the smallest span of the excerpt that completely and efficiently answers the question.

We then present the excerpt as a user prompt to the model and collect the generated QA pairs. Here are two example generated QA pairs from the `Knowledge` set of NEWS:

> **QA Pair Generated by GPT-4: Example #1**
>
> **Excerpt (User prompt):** ...According to the Stockholm International Peace Research Institute (SIPRI), the US accounted for 69% of Israel's arms imports between 2019 and 2023...
> **Question:** According to the Stockholm International Peace Research Institute (SIPRI), what percentage of Israel's arms imports between 2019 and 2023 came from the US?
> **Answer:** 69%

> **QA Pair Generated by GPT-4: Example #2**
>
> **Excerpt (User prompt):** ...Wednesday's event will be moderated by tech entrepreneur David Sacks, a close ally of the Tesla founder and a supporter of Mr DeSantis...
> **Question:** Who will moderate Wednesday's Twitter Spaces event featuring Mr DeSantis?
> **Answer:** tech entrepreneur David Sacks

## D.2 DATASET SEGMENTATION

Table 7 shows examples from **MUSE** and Table 8 presents detailed statistics for **MUSE**. For both the NEWS and BOOKS datasets, we include the type of documents along with the number of tokens in each dataset. Additionally, **MUSE** incorporates $\mathcal{D}_{\text{retain}}^{(\text{reg})}$, a distinct retain set which is seen by $f_{\text{target}}$ but not included in $\mathcal{D}_{\text{forget}}$. This set is used exclusively with the GDR and KLR regularizers discussed.

To ensure that regularized methods do not directly optimize towards the evaluation set $\mathcal{D}_{\text{retain}}$, $\mathcal{D}_{\text{retain}}^{(\text{reg})}$ is kept disjoint from $\mathcal{D}_{\text{retain}}$.

Table 7: Examples of **MUSE**. Each corpus has `Verbatim` text and `Knowledge` sets (QA pairs derived from the original text) for evaluating verbatim and knowledge memorization. In NEWS, $\mathcal{D}_{\text{forget}}$ and $\mathcal{D}_{\text{retain}}$ are two disjoint sets of news articles. In BOOKS, $\mathcal{D}_{\text{forget}}$ is the Harry Potter book series while $\mathcal{D}_{\text{retain}}$ consists of wiki articles about the series. The sizes of the forget and retain sets are reported in tokens in (). Note that only the `Verbatim` texts within the Forget Set are included in our training data, while all `Knowledge` sets (QA pairs) serve for evaluations.

| Corpus | Forget Set | Retain Set |
|--------|-----------|-----------|
| | **NEWS ARTICLE** (0.8 M tokens) | **NEWS ARTICLE** (1.6 M tokens) |
| NEWS | `MP Stuart McDonald has been appointed as the SNP's new treasurer` | `A father whose 12-year-old son was killed by an IRA bomb 30 years ago` |
| | **Q**: What position has Stuart McDonald MP been appointed to? **A**: The SNP's new treasurer | **Q**: Who was affected by the IRA bomb 30 years ago? **A**: A father whose 12-year-old son |
| | **HARRY POTTER BOOKS** (1.1 M tokens) | **HARRY POTTER FANWIKI** (0.5 M tokens) |
| BOOKS | `"There's more in the frying pan," said Aunt Petunia, turning eyes on her massive son.` | `This page contains a list of spells: Portuguese for 'open'.` |
| | **Q**: What does Aunt Petunia tell her son? **A**: There's more in the frying pan. | **Q**: What is the spell used to open things? **A**: Portuguese |

Table 8: **Statistics of the MUSE dataset.** Corpus sizes are reported in tokens, shown in (). Retain Set$_{\text{reg.}}$ is disjoint from the standard Retain Set used in evaluation and is employed in unlearning training to preserve utility through regularizers.

| Corpus | Forget Set | Retain Set | Retain Set$_{\text{reg.}}$ | Holdout Set |
|--------|-----------|-----------|------------------|-------------|
| NEWS | News Articles (3.3M) | News Articles (1.6M) | News Articles (1.6M) | News Articles (2.0M) |
| BOOKS | Harry Potter Books (1.1M) | Harry Potter FanWiki (0.5M) | Harry Potter FanWiki (0.2M) | Harry Potter Books (0.6M) |

