# OpenReview forum: "MUSE: Machine Unlearning Six-Way Evaluation for Language Models"
_ICLR.cc/2025/Conference — ICLR 2025 Poster_

### Official Review · Reviewer_pArk · 2024-10-22

**Soundness:** 3
**Presentation:** 3
**Contribution:** 2
**Rating:** 5
**Confidence:** 3

**Summary:**

This paper introduces a comprehensive benchmark to assess LMs' performance after unlearning according to six diverse desirable properties for unlearned models, including verbatim memorization, QA memorization, MIA performance, utility, scalability, and sequential unlearning requests. The paper considers two Corpus including NEWS and BOOKS, to evaluate the eight existing popular unlearning algorithms. Comprehensive experiments are conducted to reveal the effectiveness and limitations of existing methods.

**Strengths:**

1. The paper is well-written and easy to follow.

2. Compared to previous works, the paper identifies 6 desirable properties for unlearning and formulates them as evaluation metrics.

3. The paper covers a wide range of existing unlearning methods and conducts comprehensive evaluations with the proposed evaluation metrics.

**Weaknesses:**

1. My major concern is that the experimental results are not well explained. For example, Task Vector's performance in Table 3 is somehow neglected. The results of Task Vector seem inconsistent with prior works such as [1]. More celebrations should be included to discuss why task vector fails on unlearning.

2. Though the paper proposes six properties. These properties may have already been discussed in related works. From my perspective, only C6 sustainability is the new criteria for evaluation. C3, which compares MIA scores between unlearned models and retrained models, is not sound for under-unlearning.

3. In terms of C3, which states that 'Unlearning leads to privacy leakage,' under-unlearning (PrivLeak < 0) may not be regarded as privacy leakage. I agree that the perfect unlearning should let PrivLeak close to 0. However, in practice, it is not likely for the adversary to compare unlearned models with retrained models. Instead, the adversary only calculates the MIA for the unlearned models to determine membership. Therefore, under-unlearning should be one feasible outcome.






[1] Liu, Zheyuan, et al. "Towards safer large language models through machine unlearning." arXiv preprint arXiv:2402.10058 (2024).

**Questions:**

Please refer to my weaknesses. Also, I have one more question.


1. Figure 2 seems confusing. Why should the data distribution of 'forget' be close to 'holdout'? This is not reflected in the proposed PrivLeak metric.

---

> ### Author Response · Authors · 2024-11-25
>
> We thank the reviewer for their thoughtful feedback and for recognizing our comprehensive experimental evaluation and thoroughness. Below, we provide responses to the reviewer’s concerns.
>
> ## Response to Weaknesses and Questions
>
> - **My major concern is that the experimental results are not well explained. For example, Task Vector's performance in Table 3 is somehow neglected. The results of Task Vector seem inconsistent with prior works such as [1].**
>
> Our experimental results in Table 3 show that Task Vector performs poorly in preventing verbatim memorization, knowledge memorization, and privacy leakage on both datasets. This finding aligns with the conclusions drawn in [1], where the authors explicitly mentions that "Task Vector is not enough to remove all harmful knowledge from the model." Their analysis in Table 1 positions Task Vector as one of the less effective approaches, ranking 4th and 5th among six evaluated unlearning methods. Thank you for this feedback regarding the discussion of experimental results. We will expand our discussion of these findings in the revised version.
>
> - **Though the paper proposes six properties. These properties may have already been discussed in related works. From my perspective, only C6 sustainability is the new criteria for evaluation.**
>
> We design our evaluation with an **application-first approach**, identifying practical scenarios where unlearning may be applicable (e.g., privacy and copyright protection). From these scenarios, we derive evaluation metrics that reflect the perspectives of both data owners and model deployers. While we acknowledge some overlap with prior works, our primary contribution lies in tying these metrics to real-world applications and demonstrating their use with large-scale datasets, such as our news and books datasets.
> Furthermore, we introduce **practical evaluation scenarios** that better reflect real-world challenges. For instance, we evaluate unlearning by removing *Harry Potter* books while retaining knowledge of its **Fan Wiki**.
> To ensure reproducibility and facilitate further research, we have **open-sourced our entire codebase**, including reimplementations of unlearning methods and the evaluation framework.
>
> - **C3, which compares MIA scores between unlearned models and retrained models, is not sound for under-unlearning.**
>
> We respectfully disagree with this point regarding under-unlearning. Even without access to retrained models, under-unlearning can compromise privacy: when it occurs, an adversary can detect membership information by observing that MIA scores for $D_{forget}$ samples align more closely with $D_{train}$ patterns than with $D_{holdout}$ patterns—indicating that $D_{forget}$ samples were part of the training data. This privacy leakage problem exists independently of whether retrained models are available for comparison. In our evaluation framework, we use retrained models primarily as calibration.
>
> Furthermore, using MIAs to evaluate unlearning has been widely adopted by prior work [1] and the NeurIPS 2023 Machine Unlearning Challenge [2]. Our benchmark follows these prior standards for evaluating machine unlearning.
>
> [1] Inexact Unlearning Needs More Careful Evaluations to Avoid a False Sense of Privacy. Jamie Hayes, et al.
> [2] NeurIPS 2023 Machine Unlearning Challenge, https://unlearning-challenge.github.io/

---

> > ### Comment · Reviewer_pArk · 2024-11-26
> > **Rebuttal Acknowledgement**
> >
> > Thanks for your clarifications. Though prior works mentioned that "Task Vector is not enough to remove all harmful knowledge from the model," the performance of task vector is still effective somehow. However, in your reported results in Table 3, the Task Vector performs much worse and has similar results as f_target. I think more experiments or analyses should be conducted to show why the results differ.
> >
> > Regarding C3, I agree that using MIAs to evaluate unlearning is a good way to evaluate unlearning. Thank you for your explanations for the under-unlearning.
> >
> > In summary, I prefer to keep my review score unchanged, as I am skeptical about the task vectors' bad performance for unlearning.

---

> ### Author Response · Authors · 2024-11-26
>
> Dear Reviewer,
>
> We'd appreciate it if you'd let us know if our response has addressed your concerns.
>
> Thanks!

---

### Official Review · Reviewer_BEBK · 2024-10-28

**Soundness:** 3
**Presentation:** 3
**Contribution:** 3
**Rating:** 6
**Confidence:** 3

**Summary:**

This paper presents Muse, a benchmark designed to evaluate machine unlearning algorithms. The authors construct the evaluation corpus using content from BBC News articles and popular books, such as Harry Potter. They propose six desired metrics to assess machine unlearning performance, covering aspects such as memorization, utility, privacy, and scalability. The authors evaluate eight different unlearning algorithms using this benchmark. Experimental results indicate that, while current methods achieve effective unlearning, they also lead to substantial drops in utility and privacy.

**Strengths:**

- This paper overall is well-written and easy to follow.
- The paper offers a more comprehensive benchmark to evaluate machine unlearning, and conduct thorough experimental evaluation and analysis on several existing algorithms on all six metrics.

**Weaknesses:**

- This work only evaluate four families of machine unlearning methods, which are all training-based. However, training-free approaches, such as those via model editing (e.g. arxiv:2202.05262) or weight pruning (e.g. arxiv:2403.01267), are also relevant and are not covered here. The authors should either incorporate these methods or explain why they were excluded.
- The authors propose six desired properties for machine unlearning, with three reflecting the data owner’s perspective and three for the model deployer. While these criteria are generally reasonable, it remains unclear if they reflect real-world practitioner concerns. If these are indeed practitioner-driven priorities, the authors should provide references or practitioner insights to support why these metrics were chosen over others not included.

**Questions:**

In Table 5 of Appendix B.4, the authors report GPU times for each unlearning method, with the baseline retrain method requiring 184,320 hours, which seems unusually high. In the current setup, retraining starts with a base model, and the retrain model is finetuned on this base model for 5 epochs. I understand the authors' rationale that, in practical applications, the forget set is integrated into the entire pretrained corpus, making full-corpus retraining necessary rather than just training on the retain set. However, the authors should clarify this approach more explicitly in the experimental setup and justify why finetuning only on the retain set is a valid simulation of full-corpus retraining.

---

> ### Author Response · Authors · 2024-11-25
>
> We thank the reviewer for their thoughtful feedback and for recognizing our comprehensive experimental evaluation. Below, we provide responses to the reviewer’s concerns.
>
> ## Response to Weaknesses and Questions
>
> - **Evaluating Additional Unlearning Methods (e.g., Weight Pruning and Model Editing)**
>
> Thank you for suggesting additional baselines. In response, we have extended our evaluation to include the **selective weight pruning method**, with results summarized below:
>
> | Method                  | C1. No Verbatim Mem. | C2. No Knowledge Mem. | C3. No Privacy Leak. | C4. Utility Preserv. |
> |-------------------------|-----------------------|------------------------|----------------------|----------------------|
> | Selective Pruning   | 32.1                 | 55.3                  | 61.6                | 48.2                |
>
> The results demonstrate that while selective pruning achieves reasonable utility preservation, its effectiveness in removing memorized information is limited.
>
> Regarding **model editing**, its primary goal is to modify a model’s behavior for specific input-output pairs, whereas machine unlearning focuses on completely removing a training sample's influence. For example, model editing might reassign the prediction of (x, y) to a target output \( y' \), while unlearning ensures the model behaves as if (x, y) was never part of the training data. Given these distinct objectives, model editing methods are not directly applicable to the unlearning problem. However, we are happy to incorporate any other unlearning methods the reviewer may suggest.

---

> > ### Author Response · Authors · 2024-11-25
> >
> > - **Relevance of Evaluation Metrics to Real-World Practitioner Concerns**
> >
> > By constructing an evaluation framework that provides **multi-dimensional metrics** motivated from real world priorities from two key stakeholder perspectives, we provide a tool for profiling different unlearning algorithms so that different system providers can make algorithmic decisions based on their priority. In the following, we provide some examples which hopefully could clarify why our choice of metrics are motivated from real world scenarios rather than an arbitrarily chosen list.
> >
> > - **C1. No Verbatim Memorization:**
> > Consider a system provider that deals with unlearning requests related to memorization of sensitive personal information (e.g. social security number). A strong guarantee of reduced chance of model outputting the social security number after unlearning should be one of the top priority when choosing unlearning algorithms.
> >
> > - **C2. No Knowledge Memorization:**
> > Consider a different system that is designed to help people in creative processes (e.g. story writing). Then personal information leakage may not be a main concern, but knowledge memorization may be important when facing copyright related unlearning requests. For example, if an author found that the system is creating stories based on their work X, they may submit a request based on copyright infringement, and the system provider would want the model to no longer use knowledge from the work X after unlearning.
> >
> > - **C3. No Privacy Leakage:**
> > Consider yet another different system that is deployed in hospitals, with potential access to medical records and diagnosis. In this case, a much higher privacy requirement is generally needed than other typical cases for unlearning. Because in general whether an input was used to train the model may not be a top secret, but in this scenario membership inference may lead to direct linkage of a certain kind of diseases to a certain people, which may be an undesirable outcome in many cases.
> >
> > - **C4. Utility Preservation:**
> > This is a fundamental metric that needs to be considered in most cases.
> >
> > - **C5. Scalability (Deletion Capacity):**
> >   Scalability, also known as deletion capacity, is critical for systems that need to handle varying sizes of unlearning requests. Depending on the nature of the data and user bases, scalability could be high or low priority. For the example in C1, the unlearning requests related to personal information leakage are likely short, but for the example in C2, the unlearning requests may be much larger (e.g. a book or several books). The importance of deletion capacity has been emphasized in prior theoretical works [1, 2, 3].
> >
> > - **C6. Sustainability:**
> > In some systems, such as the example in C3, unlearning requests may happen relatively infrequently but each time it happens, it is related to serious privacy concerns. As a result, it is not ideal to wait and batch many unlearning requests together, so the system needs to have the capability to handle a series of unlearning requests. In this case, this dimension of metrics would be very important for choosing the right unlearning algorithm to deploy.
> >
> >
> > Furthermore The Amazon Unlearning Challenge (https://llmunlearningsemeval2025.github.io/) underscores the relevance of our criteria by including No Memorization (aligned with C1 & C2) and Membership Inference Attacks (aligned with C3) as key metrics. Metrics like C6 (sustainability) and C5 (scalability) are also important in real-world applications. For example, C6 aligns with regulatory requirements like GDPR’s “right to be forgotten,” mandating timely data deletion. Scalability evaluates how algorithms manage varying sizes of forget sets. Our experiments (Figure 6) reveal that methods like NPO perform well with small forget sets but degrade significantly with larger sets, providing critical insights for developers.
> >
> > [1] Remember What You Want to Forget: Algorithms for Machine Unlearning, NeurIPS 2021
> > [2] Algorithms that Approximate Data Removal: New Results and Limitations, NeurIPS 2022
> > [3] Certified Minimax Unlearning with Generalization Rates and Deletion Capacity, NeurIPS 2023

---

> ### Author Response · Authors · 2024-11-26
>
> Dear Reviewer,
>
> We'd appreciate it if you'd let us know if our response has addressed your concerns.
>
> Thanks!

---

> > ### Comment · Reviewer_BEBK · 2024-11-26
> >
> > Thanks you for your responses. I will maintain my current score and recommend the paper for acceptance.

---

### Official Review · Reviewer_hTCm · 2024-10-31

**Soundness:** 2
**Presentation:** 3
**Contribution:** 4
**Rating:** 8
**Confidence:** 4

**Summary:**

The paper proposes a systematic evaluation framework called MUSE to assess unlearning algorithms for LLMs. They consider 6 dimensions in their evaluation setup, from both the perspectives of data owners and model deployers. The 6 dimensions include verbatim memorization, knowledge memorization, privacy leakage, utility preservation, scalability wrt size of removal requests and sustainability over sequential unlearning requests. They apply MUSE to evaluate eight representative unlearning algorithms on two datasets, finding that current methods struggle to meet the full set of ideal requirements, especially in preventing privacy leakage.

**Strengths:**

- What the authors propose is very helpful for the community. Plenty of work is focused on developing approximate unlearning methods for LLMs, and evaluation methods employed are all too often ad-hoc rather than comprehensive and rigorous. They setup authors propose is well-thought through and covers all meaningful dimensions (at least that I see).
- Thorough analysis of unlearning algorithms, for 2 datasets
- I particularly like the realistic setup for utility preservation of Harry Potter.

**Weaknesses:**

(more see questions)

- Only evaluating one LLM, in one finetuning regime.
- No justification for the high MIA performances, which is needed to evaluate how realistic the setup and its conclusions are.
- Limited utility evaluation.
- Minor clarifications needed

**Questions:**

- I'd like to understand the MIA a bit more. You mention that the AUC of the MIA against f_target is 0.0. If I understand it correctly, this refers to the very classic MIA where you have to distinguish between members (D_train) and non-members (D_holdout), without any unlearning. Hence two questions,
	- Why do you not adapt the standard notion of MIA AUC where the 1 label corresponds to membership (which would mean the AUC on f_target would be close to 1.0 rather than 0.0). I think this would make it more clear for the reader.
	- Why does it perform so well? While the initial work of Shi et al. indeed shows great performances for Min-k% Prob, it is by now well known that the great performance was due to a distribution shift rather than measuring memorization and that typical MIAs against LLMs do not work much better than a random guess [1,2,3]. I don't believe your setup has a distribution shift (as the AUC for f_retrain is 0.47), yet I wonder why the attack works near-perfectly while all MIAs against LLMs in proper setups . Could you add experiments that convincingly show why it works so well in your setup compared to others? My intuition would be that it is due to the fact that you use 5 epochs for finetuning. Could you add ablations to your unlearning evaluation for less epochs? This would help understand how your methods and conclusions apply to (more realistic) setups where LLMs memorize less.

- As all evaluation for metrics C4-6 are based on utility, I wonder if the metric used for utility should be more elaborate. Right now, you only measure the knowledge memorization on the custom created question-answering pairs. But I think it would be meaningful to also include e.g. perplexity on D_retain or even performances on well known benchmarks (this would also help clarify whether the finetuning represents a realistic setup for a useful f_target). Could you add this as well?
- Clarifications
	- Could you confirm in Sec 3.2 where the holdout set for books come from? Is it a random split of the full books or is it data from the harry potter wiki?
	- Maybe nit, but I was quite confused by the notion of f_retrain. The first time you mention f_retrain is in Sec 3.1 if I'm not mistaken. Could you formalize what this refers to in Sec 2? Otherwise it's hard to follow. And it might also be worth it to rename either 'retrain' in f_retrain or 'retain' in D_retain, as I found this to be confusing too;

**I believe the framework the paper proposes is very valuable, so I'm generally in favor of accepting the paper. However I also believe my concerns are valid and clarifying them (they should be very fixable) would improve the work. Hence, i'm sticking with a 5 right now, but happy to change the score when my concerns are properly addressed during the rebuttal.**

[1] Duan, M., Suri, A., Mireshghallah, N., Min, S., Shi, W., Zettlemoyer, L., ... & Hajishirzi, H. (2024). Do membership inference attacks work on large language models?. arXiv preprint arXiv:2402.07841.

[2] Meeus, M., Shilov, I.,  Jain, S., Faysse, M., Rei, M., & de Montjoye, Y. A. (2024). SoK: Membership Inference Attacks on LLMs are Rushing Nowhere (and How to Fix It). arXiv preprint arXiv:2406.17975.

[3] Das, D., Zhang, J., & Tramèr, F. (2024). Blind baselines beat membership inference attacks for foundation models. arXiv preprint arXiv:2406.16201.

---

> ### Author Response · Authors · 2024-11-25
>
> We thank the reviewer for acknowledging the realistic utility preservation setup for Harry Potter and the comprehensiveness of our evaluation framework. Below, we provide responses to the reviewer’s concerns.
>
> ### **Response to weaknesses and questions**
>
> - **High MIA performance requires justification. AUC of MIA against f_target is reported as 0.0—should it be flipped?**
> Yes, this refers to the classic MIA distinguishing between members (D_train) and non-members (D_holdout). An AUC near 0.5 indicates an ineffective attack, while values near 0 or 1 indicate a strong attack. You’re right about flipping the labels for clarity. We’ll update the paper accordingly, as this won’t affect our results but will improve clarity. Thank you for the helpful feedback!
>
> - **Why do MIAs perform near-perfectly in your setup compared to others? Can you add ablations for fewer fine-tuning epochs?**
> As you correctly pointed out, our setup does not have a distribution shift. To address your question, we conducted experiments to examine the impact of finetuning epochs on MIA performance. Specifically, we evaluated the AUC scores of the f_target model on news data fintuned for varying numbers of epochs. Additionally, as per your recommendation above, we flipped the labels of examples in D_train and D_holdout and reported the results below:
>     | Method  | Epoch 1 | Epoch 2 | Epoch 5 |
>     | ------- | ------- | ------- | ------- |
>     | Target  | 0.97    | 0.99    | 1.00    |
>     | Retrain | 0.52    | 0.52    | 0.52    |
>     | GA      | 0.49    | 0.50    | 0.50    |
>     | GA_GDR  | 0.13    | 0.06    | 0.01    |
>     | GA_KLR  | 0.89    | 0.94    | 0.98    |
>     | NPO     | 0.47    | 0.44    | 0.41    |
>     | NPO_GDR | 0.14    | 0.08    | 0.02    |
>     | NPO_KLR | 0.93    | 0.97    | 0.98    |
> These results suggest that the MIA method performs well in our setup with different number of epochs. We are also happy to evaluate any additional MIA methods that the reviewer recommends and include the results in the paper.
>
>
> - **Metrics C4-6 are utility-based—should the utility metric be more elaborate?**
> Thank you for your suggestion. We have incorporated additional evaluations to address your concern. Specifically, we added perplexity measurements on D_retain and the performance on HellaSwag for the target model finetuned on news. The updated results are summarized below, and we will include these experiments in the revised paper:
> | Method | KnowMem on D_retain (EM) | PPL on D_retain | HellaSwag (Acc %) |
> |--------|-------------------------|-----------------|-------------------|
> | Target | 55.2 | 7.63 | 72.5 |
> | Retrain | 55.0 | 7.72 | 73.2 |
> | GA | 0.0 | 26.1 | 38.2 |
> | GA_GDR | 27.3 | 17.5 | 42.5 |
> | GA_KLR | 44.8 | 9.4 | 52.4 |
> | NPO | 0.0 | 30.2 | 34.1 |
> | NPO_GDR | 40.5 | 12.2 | 55.3 |
> | NPO_KLR | 45.4 | 9.8 | 48.2 |
>
> - **Clarify the source of the holdout set for Harry Potter books.**
> The holdout set for books is a split of the Harry Potter books, as shown in Table 8.
>
> - **f_retrain definition and terminology overlap with D_retain is confusing.**
> Thank you for pointing this out. We apologize for the typo in line 125 of Section 2. It should be: *“Exact unlearning f_unlearn ensures f_unlearn is behaviorally identical to the model resulting from retraining from scratch, denoted f_retrain.”*
> To address the confusion, we will rename f_retrain to f_oracle in the final version for greater clarity and to avoid overlap with D_retain. Thank you for the suggestion!

---

> > ### Comment · Reviewer_hTCm · 2024-11-25
> > **Answer to rebuttal**
> >
> > Many thanks for the elaborate rebuttal. The comments addressed my concerns and I believe the edits and additional experiments will improve the paper. I just upgraded my score to an 8 to reflect it, hope it gets accepted.

---

### Official Review · Reviewer_Kopk · 2024-11-03

**Soundness:** 3
**Presentation:** 3
**Contribution:** 2
**Rating:** 6
**Confidence:** 3

**Summary:**

This article introduces a method that provides an evaluation framework MUSE for machine unlearning. This evaluation framework assesses the machine unlearning capability from different perspectives through six types of evaluations. Experiments conducted on a series of 7B parameter models validate the framework, comparing different methods and their effectiveness.

**Strengths:**

1: This paper provides a comprehensive and detailed study of methods for machine unlearning, conducting an in-depth evaluation from six perspectives. It offers a thorough assessment framework covering aspects such as semantics, continuity, knowledge, memory, and privacy. Compared to previous evaluation frameworks, this approach has a broader scope, assesses from more perspectives, and utilizes a larger dataset, demonstrating the framework's comprehensiveness and effectiveness.

2: This paper provides a detailed introduction to each method of unlearning, explaining the corresponding implementation processes with comprehensive formulas and theoretical explanations.

3: The experiments on different datasets are presented clearly, with comparisons across various categories.

4: What I appreciate most is the authors’ effort in providing evaluation metrics, giving clarification for each method with straightforward assessments. Although some metrics, such as C5 and C6, are not as clearly defined, the authors’ attempts to evaluate unlearning across various dimensions are commendable.

**Weaknesses:**

1: Although the authors provide numerous metrics, many of them heavily rely on previous methods. For example, the C3 metric for privacy assessment has already been addressed in a series of earlier approaches. Additionally, it remains unclear whether certain metrics, such as C5 and C6, are truly essential for evaluating machine unlearning, as their explanations in the paper are not entirely clear.

While I appreciate the advantages noted in Strengths, I would like to see more about how these various methods are integrated within this framework and what specific improvements it offers over prior approaches. Metrics like C1, C2, and C3, for instance, have elements originating from previous work. Although Table 1 highlights the advantages of this framework over the previous TOFU metrics, I believe these advantages are not significant and merely provide comparisons with a few additional metrics.

2: The machine unlearning methods presented in Section 4 of this paper all adopt previous approaches, which makes me wonder if this part could be incorporated into the experimental section instead. It occupies a considerable amount of space, and the methods discussed are largely drawn from prior work. The overall structure of the paper could benefit from further refinement to enhance its clarity and conciseness.

3: Although the paper conducts numerous experiments and evaluates them across various metrics, the results intended to be conveyed by these different metrics are somewhat unclear. For example, if we introduce a set of metrics (C1 to C6) to evaluate different machine unlearning approaches, it raises several questions: Why were these metrics chosen? What conclusions do the experiments under these metrics yield? What consistencies or insights do they offer? The paper lacks sufficient commentary in this regard, and its quality could be enhanced by incorporating more examples and providing clearer justifications for the experiments, thus improving the interpretation of the findings.

**Questions:**

For me, I would like to see a more detailed explanation of the role each metric plays within the framework used, as well as how the experiments impact these metrics across different methods. This section of the discussion could be further refined and deepened to provide clearer insights into the significance of each metric and the implications of the experimental results.

---

> ### Author Response · Authors · 2024-11-25
>
> We thank the reviewer for appreciating the comprehensiveness of our evaluation framework and experimental analysis. Below, we provide responses to the reviewer’s concerns.
>
> ## Response to Weaknesses and Questions
>
> - **Necessity of Metrics Such as C5 (Scalability) and C6 (Sustainability)**
>    C6 (sustainability) is important in real-world scenarios, particularly for compliance with regulations such as GDPR’s “right to be forgotten,” which mandates the timely processing of data removal requests. For example, when requests are submitted 31 days apart, they cannot be batched together, requiring efficient handling of sequential unlearning tasks.
>    Similarly, C5 (scalability) also known as deletion capacity is crucial for understanding how algorithms perform with varying sizes of forget sets and request frequencies. For instance, our experiments (Figure 6) demonstrate that methods like NPO perform well with small forget sets but degrade significantly with larger ones. These trends inform system developers on selecting unlearning algorithms that align with their operational needs.
>
> - **Integration of Framework and Improvements Over Prior Work**
>    The key contribution of our framework is not merely adding metrics but selecting relevant ones to create a **unified view** that balances the perspectives of data owners and service providers. Unlike prior work, which often focused on theoretical or synthetic evaluations, our **application-first approach** tailors the evaluation to real-world scenarios, such as privacy protection and copyright compliance. For example, we evaluate unlearning by removing *Harry Potter* books while retaining knowledge of related fan wikis, providing a more practical evaluation scenario compared to TOFU’s synthetic data-based evaluations.  Additionally, we have **open-sourced our entire codebase**, including reimplementations of unlearning methods and the evaluation framework, to facilitate reproducibility and further research.

---

> ### Author Response · Authors · 2024-11-25
>
> - **Clarifying the Role and Rationale for Metrics**
> By constructing an evaluation framework that provides **multi-dimensional metrics** motivated from real world priorities from two key stakeholder perspectives, we provide a tool for profiling different unlearning algorithms so that different system providers can make algorithmic decisions based on their priority. In the following, we provide some examples which hopefully could clarify why our choice of metrics are motivated from real world scenarios rather than an arbitrarily chosen list.
>
> **C1. No Verbatim Memorization:**
> Consider a system provider that deals with unlearning requests related to memorization of sensitive personal information (e.g. social security number). A strong guarantee of reduced chance of model outputting the social security number after unlearning should be one of the top priority when choosing unlearning algorithms.
>
> **C2. No Knowledge Memorization:**
> Consider a different system that is designed to help people in creative processes (e.g. story writing). Then personal information leakage may not be a main concern, but knowledge memorization may be important when facing copyright related unlearning requests. For example, if an author found that the system is creating stories based on their work X, they may submit a request based on copyright infringement, and the system provider would want the model to no longer use knowledge from the work X after unlearning.
>
> **C3. No Privacy Leakage:**
> Consider yet another different system that is deployed in hospitals, with potential access to medical records and diagnosis. In this case, a much higher privacy requirement is generally needed than other typical cases for unlearning. Because in general whether an input was used to train the model may not be a top secret, but in this scenario membership inference may lead to direct linkage of a certain kind of diseases to a certain people, which may be an undesirable outcome in many cases.
>
> **C4. Utility Preservation:**
> This is a fundamental metric that needs to be considered in most cases.
>
> **C5. Scalability**
> Depending on the nature of the data and user bases, scalability could be high or low priority. For the example in C1, the unlearning requests related to personal information leakage are likely short, but for the example in C2, the unlearning requests may be much larger (e.g. a book or several books).
>
> **C6. Sustainability:**
> In some systems, such as the example in C3, unlearning requests may happen relatively infrequently but each time it happens, it is related to serious privacy concerns. As a result, it is not ideal to wait and batch many unlearning requests together, so the system needs to have the capability to handle a series of unlearning requests. In this case, this dimension of metrics would be very important for choosing the right unlearning algorithm to deploy.
>
> In summary, our framework provides a multi-dimensional view into an unlearning algorithm, motivated from real world needs from both the system provider and data owner’s perspective. This provides a clear profile for balancing different priorities when choosing the right algorithm to deploy. Furthermore, we hope this framework could also guide future work on designing specialized unlearning algorithms that are especially strong at certain dimensions, for targeting specific application scenarios. Thank you for your valuable suggestion; we will include a detailed discussion in the revised version.
>
>
> - **Incorporating Machine Unlearning Methods into the Experimental Section**
> Thank you for the suggestion. We will move the discussion of machine unlearning methods in Section 4 to the experimental section in the revised version

---

> > ### Author Response · Authors · 2024-11-26
> >
> > Dear Reviewer,
> >
> > We'd appreciate it if you'd let us know if our response has addressed your concerns.
> >
> > Thanks!

---

> ### Comment · Reviewer_Kopk · 2024-11-26
> **Resonse to Authors**
>
> Thanks for your detailed information, as the content shows the importance for future unlearning benchmark. I will increase my score to 6. I will highly recommend that authors would more focus on the reason that why the six domains are all important, because other indicators could show similar impact on the machine unlearning scenarios.

---

### Official Review · Reviewer_f19H · 2024-11-05

**Soundness:** 3
**Presentation:** 4
**Contribution:** 4
**Rating:** 6
**Confidence:** 4

**Summary:**

This paper proposes MUSE as a benchmark that evaluates unlearned models from multiple angles. The authors leverage this benchmark to test 8 popular unlearning algorithms on LLM, exposing the privacy leakage and utility degradation issues in existing unlearning methods.

**Strengths:**

This paper tackles very important problems with solid efforts to build the benchmark. The six perspectives of the benchmark are impactful and clear. I enjoy reading this work and I am convinced by the experiments, which are sufficiently comprehensive and well-designed.

**Weaknesses:**

Overall the weaknesses are not significant. I think the scale of data and number of methods may be further extended, for example, the conclusion of a method's effectiveness may change when forget set gets larger.

Also the hyperparameter tuning may be sub-optimal and require more elaboration.

Figure 4 multiple lines are using the same color which is confusing.
The blue curve in Figure 6 seems completely covered.

Minor: Line 414 GA should be GA_GDR?

**Questions:**

Although my questions do not affect my evaluation, I am curious whether the authors can comment on the method in https://arxiv.org/pdf/2410.22086? In your Figure 5, no methods are recognized as "desired", i.e. high utility and low-to-medium memorization, yet NGDiff seems to be close to desired.

---

> ### Author Response · Authors · 2024-11-25
>
> We thank the reviewer for acknowledging that our evaluation is comprehensive and well-designed. Below, we provide responses to the reviewer’s concerns.
>
> ## Response to Weaknesses and Questions
>
> - **Overall the weaknesses are not significant. I think the scale of data and number of methods may be further extended, for example, the conclusion of a method's effectiveness may change when the forget set gets larger.**
>
> **Number of Methods:**
> We appreciate the reviewer's suggestion on extending the evaluation to include more methods. In the next version, we will incorporate recently published unlearning methods, including:
> [1] *Heterogeneous Decentralized Machine Unlearning with Seed Model Distillation*
> [2] *Machine Unlearning by Reversing the Continual Learning*
>
> Additionally, we have open-sourced the entire codebase and set up an easy-to-use leaderboard. This allows method developers to contribute their own implementations, enabling evaluation of any new unlearning approaches on our benchmark.
>
> **Scale of Data:**
> Our study evaluates unlearning methods across varying sizes of forget sets, ranging from 0.8M to 3.3M tokens (Line 432). This scale is significantly larger than those used in prior evaluation benchmarks. Figure 6 in our paper analyzes the scaling trends of different methods, providing insights into how effectiveness varies with the size of the forget set.
>
> - **Also the hyperparameter tuning may be sub-optimal and require more elaboration.**
>
> As discussed in Line 1001, we use a small validation set to identify the optimal hyperparameters for each unlearning method. We will provide additional details about our hyperparameter tuning process in the revised version to ensure transparency and reproducibility.

---

> > ### Comment · Reviewer_f19H · 2024-11-27
> >
> > I have read the response but most of my concerns are not answered. The readability of Figure 4 and Figure 6 is still poor, curves are covered and uninterpretable. Line 414 is not clarified. My question about NGDiff is ignored.

---

### Author Response · Authors · 2024-11-25
**Response to the Common Questions**

We thank the reviewers for their detailed and thoughtful feedback. We appreciate the acknowledgment of our comprehensive evaluation framework and thorough experimental analysis. Below, we address the common questions and concerns raised by the reviewers.

## 1. Comparison with Prior Work

We design our evaluation with an **application-first approach**, identifying practical scenarios where unlearning may be applicable (e.g., privacy and copyright protection). From these scenarios, we derive evaluation metrics that reflect the perspectives of both data owners and model deployers. While we acknowledge some overlap with prior works, our primary contribution lies in tying these metrics to real-world applications and demonstrating their use with large-scale datasets, such as our news and books datasets.
Furthermore, we introduce **practical evaluation scenarios** that better reflect real-world challenges. For instance, we evaluate unlearning by removing *Harry Potter* books while retaining knowledge of its **Fan Wiki**.
To ensure reproducibility and facilitate further research, we have **open-sourced our entire codebase**, including reimplementations of unlearning methods and the evaluation framework.

---

## 2. Clarifying the Role and Rationale for Each Metric

By constructing an evaluation framework that provides **multi-dimensional metrics** motivated from real world priorities from two key stakeholder perspectives, we provide a tool for profiling different unlearning algorithms so that different system providers can make algorithmic decisions based on their priority. In the following, we provide some examples which hopefully could clarify why our choice of metrics are motivated from real world scenarios rather than an arbitrarily chosen list.


- **C1. No Verbatim Memorization:**
Consider a system provider that deals with unlearning requests related to memorization of sensitive personal information (e.g. social security number). A strong guarantee of reduced chance of model outputting the social security number after unlearning should be one of the top priority when choosing unlearning algorithms.

- **C2. No Knowledge Memorization:**
Consider a different system that is designed to help people in creative processes (e.g. story writing). Then personal information leakage may not be a main concern, but knowledge memorization may be important when facing copyright related unlearning requests. For example, if an author found that the system is creating stories based on their work X, they may submit a request based on copyright infringement, and the system provider would want the model to no longer use knowledge from the work X after unlearning.

- **C3. No Privacy Leakage:**
Consider yet another different system that is deployed in hospitals, with potential access to medical records and diagnosis. In this case, a much higher privacy requirement is generally needed than other typical cases for unlearning. Because in general whether an input was used to train the model may not be a top secret, but in this scenario membership inference may lead to direct linkage of a certain kind of diseases to a certain people, which may be an undesirable outcome in many cases.

- **C4. Utility Preservation:**
This is a fundamental metric that needs to be considered in most cases.

- **C5. Scalability (Deletion Capacity):**
  Scalability, also known as deletion capacity, is critical for systems that need to handle varying sizes of unlearning requests. Depending on the nature of the data and user bases, scalability could be high or low priority. For the example in C1, the unlearning requests related to personal information leakage are likely short, but for the example in C2, the unlearning requests may be much larger (e.g. a book or several books). The importance of deletion capacity has been emphasized in prior theoretical works [1, 2, 3].

- **C6. Sustainability:**
In some systems, such as the example in C3, unlearning requests may happen relatively infrequently but each time it happens, it is related to serious privacy concerns. As a result, it is not ideal to wait and batch many unlearning requests together, so the system needs to have the capability to handle a series of unlearning requests. In this case, this dimension of metrics would be very important for choosing the right unlearning algorithm to deploy.

In summary, our framework offers a multi-dimensional view of unlearning algorithms, addressing real-world needs of model developers and data owners. It helps balance priorities when selecting algorithms and guides future work on specialized unlearning methods for specific applications.

[1] Remember What You Want to Forget: Algorithms for Machine Unlearning, NeurIPS 2021
[2] Algorithms that Approximate Data Removal: New Results and Limitations, NeurIPS 2022
[3] Certified Minimax Unlearning with Generalization Rates and Deletion Capacity, NeurIPS 2023

---

### Meta-Review · Area_Chair_j2YC · 2024-12-24

**Metareview:**

Authors provide a comprehensive treatment of "evaluating machine unlearning algorithms" and provide 6 evaluation criteria for such algorithms. These criteria are broadly classified into two categories pertinent to the data owners' and the deployers' interests. Paper provides an extensive evaluation of known unlearning algorithms using their criteria and uncover non-trivial characteristics of them. It is noted that some of the criteria are separately known in past literature, however this work claims to provide a unified overview through an applied lens.

**Additional Comments On Reviewer Discussion:**

All the reviewers had concerns with lack of clarity in terms of (1) justifications for the criteria, (2) interpretation of the evaluation, and (4) broad take-aways learning from it. Authors addressed some of these concerns during discussion phase, however few reviewers still had some concerns.

---

### Decision · Program_Chairs · 2025-01-22

Accept (Poster)